# What Are Large Language Models Mapping to in the Brain? A Case Against Over-Reliance on Brain Scores

## Abstract

Given the remarkable capabilities of large language models (LLMs), there has
been a growing interest in evaluating their similarity to the human brain. One
approach towards quantifying this similarity is by measuring how well a model
predicts neural signals, also called "brain score". Internal representations from
LLMs achieve state-of-the-art brain scores, leading to speculation that they share
computational principles with human language processing. This inference is only
valid if the subset of neural activity predicted by LLMs reflects core elements
of language processing. Here, we question this assumption by analyzing three
neural datasets used in an impactful study on LLM-to-brain mappings, with a
particular focus on an fMRI dataset where participants read short passages. We
first find that when using shuffled train-test splits, as done in previous studies
with these datasets, a trivial feature that encodes temporal autocorrelation not only
outperforms LLMs but also accounts for the majority of neural variance that LLMs
explain. We therefore caution against shuffled train-test splits, and use contiguous
test splits moving forward. Second, we explain the surprising result that untrained
LLMs have higher-than-expected brain scores by showing they do not account
for additional neural variance beyond two simple features: sentence length and
sentence position. This undermines evidence used to claim that the transformer
architecture biases computations to be more brain-like. Third, we find that brain
scores of trained LLMs on this dataset can largely be explained by sentence
position, sentence length, and static word vectors; a small, additional amount is
explained by sense-specific word embeddings and contextual representations of
sentence structure. We conclude that over-reliance on brain scores can lead to
over-interpretations of similarity between LLMs and brains, and emphasize the
importance of deconstructing what LLMs are mapping to in neural signals.

## 1 Introduction

Recent developments in large language models (LLMs) have led many to wonder whether LLMs
process language like humans do. Whereas LLMs acquire many abstract linguistic generalizations, it
remains unclear to what extent their internal machinery bears resemblance to the human brain [1]. A
number of studies have attempted to answer this question through the framework of neural encoding
[2–4]. Within this framework, an LLM's internal representations of some linguistic stimuli are used
to predict brain activity during comprehension of the same stimuli. Results have been uniformly
positive, showing that LLM representations are highly effective at predicting neural signals [5, 6].

In one impactful study, authors evaluated the brain scores of 43 models on three neural datasets [2].
They found that GPT2-XL [7] achieved the highest brain score and, in one neural dataset, accounted
for 100% of the "explainable" neural variance (i.e., taking into account the noise inherent in the data)

[8]. This result was interpreted as evidence that the brain may be optimizing for the same objective as GPT2, namely, next-word prediction. Surprisingly, the authors further found that untrained (i.e. randomly initialized) LLMs predict neural activity well, leading to speculations that the transformer architecture biases computations to be more brain-like. The finding that untrained LLMs predict neural signals significantly above chance has been replicated in other studies [9, 4, 10].

More generally, many studies have compared models to brain activity and concluded that high prediction performance reveals correspondence between some interesting aspect of the model and biological linguistic processing [4, 11–14]. One issue with this approach is that it assumes that the subset of neural activity predicted by a model reflects core processes of the human language system [15]. However, this assumption is not necessarily true. For example, a recent paper found that, when participants listen to stories, the fMRI signal includes an initial ramping, positional artifact [16]. It is likely that LLMs which contain absolute positional embeddings would be able to predict this ramping signal, whereas a simpler model such as a static word embedding (e.g. GloVe, [17]) would not, leading to exaggerated differences between LLMs and GloVe due to reasons of little theoretical interest. This issue relates to a more general trend in machine learning research: a complex algorithm solves a task, but it is later discovered that the key innovation was a very simple component of the algorithm [18]. Analogous to Weinberger [18], without attempting to rigorously deconstruct the mapping between LLMs and brains, it is possible to draw erroneous conclusions about the brain's mechanisms for processing language.

We analyze the same three neural datasets used in [2]. These include the Pereira fMRI dataset, where participants read short passages [8]; the Fedorenko electrocorticography (ECoG) dataset, where participants read isolated sentences [19]; and the Blank fMRI dataset, where participants listened to short stories [20]. As in Schrimpf et al. [2], we focus our analyses on the Pereira dataset. In order to deconstruct the mapping between LLMs and the brain, we follow Reddy and Wehbe [21] and de Heer et al. [22] by building a set of predictors that describe simple features of the linguistic input, and gradually add features that increase in complexity. Our goal is to find the simplest set of features which account for the greatest portion of the mapping between LLMs and brains.

## 2  Methods

### 2.1  Experimental data

For all three neural datasets, we used the same version as used by [2]. For additional details, refer to A.1.

**Pereira (fMRI):** The Pereira dataset is composed of two experiments. Experiment 1 (EXP1) consists of 96 passages each containing $4$ sentences, with $n = 9$ participants. Experiment 2 (EXP2) consists of 72 passages each consisting of 3 or 4 sentences, with $n = 6$ participants. Passages in each experiment were evenly divided into $24$ semantic categories which were not related across experiments (4 passages per category in EXP1, and 3 passages per category in EXP2). A single fMRI scan (TR) was taken after visual presentation of each sentence. Unless otherwise noted, we focus our results on voxels from within the "language network" in the main paper. EXP1 was a $384 \times 92450$ matrix (number of sentences $\times$ number of voxels) and EXP2 was a $243 \times 60100$ matrix. All analyses were conducted separately for each experiment.

**Fedorenko (ECoG):** Participants ($n = 5$) read $52$ sentences of length $8$ words. A total of 97 language-responsive electrodes were used across 5 participants: $47, 8, 9, 15$, and $18$, for participants 1 through 5, respectively. Neural activity was temporally averaged across the full presentation of each word after extracting high gamma, and the entire dataset was a $416 \times 97$ matrix.

**Blank (fMRI):** The dataset consisted of 5 participants listening to $8$ stories from the publicly available Natural Stories Corpus [23]. An fMRI scan was taken every 2 seconds, resulting in a total of 1317 TRs across the 8 stories. fMRI BOLD signals were averaged across voxels within each functional region of interest (fROI). There were 60 fROIs across all 5 participants, resulting in a $1317 \times 60$ matrix.

## 2.2 Language models

We focus our analyses on GPT2-XL [7], as it was shown to be the best-performing model on the Pereira dataset [10, 24, 2]. GPT2 is an auto-regressive transformer model, meaning that it can only attend to current and past inputs, trained on next token prediction. The XL variant has $\sim$1.5B parameters and 48 layers. We replicate some of our key findings on Pereira with RoBERTa-Large[25] (A.6). RoBERTa is a transformer model with bidirectional attention trained on masked token prediction, meaning that it can attend to past and future tokens. The large variant contains 335M parameters and 24 layers. Both GPT2 and RoBERTa use learned absolute positional embeddings, such that a unique vector corresponding to each token position is added to the input static embeddings.

## 2.3 LLM feature pooling

**Pereira:** Each sentence was fed into an LLM, with previous sentences from the same passage also fed as input. Since each fMRI scan was taken at the end of the sentence, we converted LLM token-level embeddings to sentence-level embeddings by summing across all tokens within a sentence (sum pooling). We used the sum pooling method because it is consistent with other neural encoding studies [26, 27], and it performed better than taking the representation at the last token which was done in [2] A.5.

**Fedorenko:** The current and previous tokens from within the same sentence were fed into the LLM as context. We converted LLM token-level embeddings to word embeddings, since each word has a neural response, by summing across tokens in multi-token words, and leaving single token words unmodified.

**Blank:** For each story, we fed the current and all preceding tokens up to a maximum context size of 512 tokens. As in Schrimpf et al. [2], for each TR, we took the representation of the word that was closest to being 4 seconds before the TR. For multi-token words, we took the representation of the last token of that word.

## 2.4 Banded ridge regression

We used ridge regression (linear regression with an L2 penalty) to predict activations for each voxel/electrode/fROI independently. We did not use "vanilla" ridge regression because it applies a single L2 penalty for all weights, whereas our analyses use multiple sets of distinct features. In such a case, a single penalty causes the regression will be biased against small feature spaces. Moreover, different L2 penalties are likely optimal for each feature space. To remedy this, we employed banded ridge regression which effectively allows a different L2 penalty to be applied to each feature space [28] (for further details, refer to A.2).

## 2.5 Out of sample $R^2$ metric

We define the brain score of a model as the out-of-sample $R^2$ metric ($R^2_{oos}$) [29]. $R^2_{oos}$ quantifies how much better a set of features performs at predicting held-out data compared to a model which simply predicts the mean of the training data (i.e. a regression with only an intercept term). To be precise, given mean squared error (MSE) values from a model using features $M$ and MSE values from an intercept only regression ($I$), then:

$$R^2_{oos} = 1 - \frac{MSE_M}{MSE_I}. \tag{1}$$

A positive (negative) value indicates that $M$ was more (less) helpful than predicting the mean of training data. We elected to use $R^2_{oos}$ over the standard $R^2$ because of this clear interpretation and because it is a less biased estimate of test set performance [29]. We use $R^2_{oos}$ over Pearson's correlation coefficient ($r$) because $R^2_{oos}$ can be interpreted as the fraction of variance explained, which lends more straightforwardly to estimating how much variance one feature space explains over others. Whenever averaging across voxels, we set $R^2_{oos}$ values to be non-negative to prevent differences in performance on noisy voxels/electrodes/fROIs from significantly impacting the results. We refer to $R^2_{oos}$ as $R^2$ throughout the rest of the paper for brevity, and use the notation $R^2_M$ to refer to the performance of features $M$.

## 2.6 Selection of best layer

We evaluate the $R^2$ for each LLM layer, and select the layer that performs best across voxels/electrodes/fROIs. Due to the stochastic nature of untrained LLMs, we selected the best layer for 10 random seeds and computed the average $R^2$ across seeds. When reporting the best layer, we refer to layer 0 as the input static layer, and layer 1 as the first intermediate layer.

## 2.7 Train, validation, and test folds:

For each dataset, we construct contiguous train-test splits by ensuring neural data from the same passage/sentence/story is not included in both train and test data. Due to low sample sizes, we employed a nested cross-validation procedure for each dataset (A.3). When computing $R^2$ across inner or outer folds, we pooled predictions across folds and computed a single $R^2$ as recommended by Hawinkel et al. [29]. The optimal parameters for banded regression were selected based on validation data.

We created shuffled train-test splits, as done in [2], of the same size as the contiguous train-test splits. Unless explicitly noted, all results are performed using contiguous train-test splits.

## 2.8 Correcting for decreases in test-set performance due to addition of feature spaces

It is possible for a "full" encoding model to perform worse than a "sub-model" (which consists of only a subset of the predictors) because we are evaluating performance on a held-out test set [22]. To address this problem, in some analyses we select the best performing sub-model for each voxel/electrode/fROI which includes a given feature of interest. For instance, to examine how much feature space $C$ adds onto features spaces $A$ and $B$, we select the best sub-model which includes $C$ and denote it as $A + B + C$*. More precisely, the $R^2$ of $A + B + C$* is:

$$R^2_{A+B+C*} = \max(R^2_C, R^2_{A+C}, R^2_{B+C}, R^2_{A+B+C}). \tag{2}$$

## 2.9 Orthogonal Auto-correlated Sequences Model (OASM)

To model temporal auto-correlation in neural activity, we construct a feature matrix for each dataset by (i) forming an $n$-dimensional identity matrix, where $n$ is the total number of time points in the dataset (per voxel / electrode / TR), and (ii) applying a Gaussian filter within "chunks" along the diagonal that correspond to temporally contiguous time points (i.e., within each passage in Pereira, each sentence in Fedorenko, and each story in Blank). This generates an auto-correlated sequence for each passage/sentence/story that is orthogonal to that of each other passage/sentence/story (A.7).

# 3 Pereira dataset

## 3.1 Shuffled train-test splits are severely affected by temporal auto-correlation

Prior LLM encoding studies using this dataset [24, 2, 10, 30, 11] used shuffled train-test splits. Here, we demonstrate that this approach compromises the evaluation of the neural predictivity of LLMs. First, we replicated the pattern of neural predictivity across GPT2-XL's layers reported in [2] and [24] when using shuffled splits. Using this procedure, early and late layers perform best and intermediate layers perform worst. Strikingly, when using the alternative approach of contiguous train-test splits, the opposite pattern is observed: intermediate layers perform best. Across layers, neural predictivity using the shuffled method is highly anti-correlated with neural predictivity using the contiguous method ($r = -.929$ in EXP1, $r = -.764$ in EXP2) (Fig. 1a).

Next, we hypothesized that much of what LLMs might be mapping to when using shuffled splits could be accounted for by OASM, a model which only represents within passage auto-correlation and between passage orthogonality. OASM out-performed GPT2-XL on both EXP1 and EXP2 (Fig. 1b, blue and red bars), revealing that a completely non-linguistic feature space can achieve absurdly high brain scores in the context of shuffled splits. This strongly challenges the assumption of multiple previous studies [2, 11, 10] that performance on this benchmark is an indication of a model's brain-likeness, .

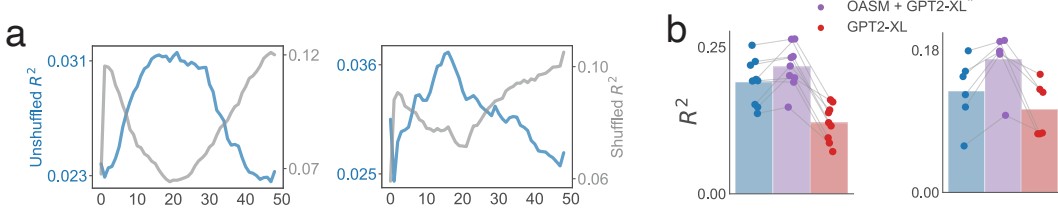

Figure 1: Comparing different approaches for creating train-test splits in the Pereira dataset. Within each panel, EXP1 results are on the left and EXP2 results are on the right (same formatting in Figure 2,3) **(a)** $R^2$ values across layers for GPT2-XL on shuffled train-test splits (gray) and contiguous (unshuffled) splits (blue). **(b)** Each dot shows the mean $R^2$ value across voxels within a participant, with bars indicating mean $R^2$ across participants.

Moreover, we find that the unique neural variance that GPT2-XL explains over OASM is very small relative to what OASM explains alone. To calculate this, we combine OASM with GPT2-XL and observe how much neural variance they explain together. To prevent OASM from ever weakening the reported performance of GPT2-XL for any voxel, we correct the $R^2$ value for each voxel with the OASM+GPT2-XL model to be at least as high as with GPT2-XL alone (denoted OASM+GPT2-XL*) (2.8). Even with these corrections, we find that $R^2_{OASM+GPT2-XL}$* was 13.6% higher than $R^2_{OASM}$ in EXP1, and 31.5% higher than $R^2_{OASM}$ in EXP2 (Fig. 1b) (% differences after averaging $R^2$ across participants). To be clear, this means that any linguistically-driven neural variance that GPT2-XL uniquely explains over OASM is far smaller (13.6% on EXP1 and 31.5% on EXP2) than what is predicted solely by OASM, a model with no linguistic features that completely lacks the ability to generalize to fully held out passages. Thus, it appears that the largest determinant of model predictivity on this dataset when using shuffled train-test splits is whether a model contains autocorrelated sequences within passages that are orthogonal between passages.

### 3.2 Untrained LLM neural predictivity is fully accounted for by sentence length and position

We next sought to deconstruct what explains the neural predictivity of untrained GPT2-XL (GPT2-XLU) in the Pereira dataset. We hypothesized that $R^2_{GPT2-XLU}$ could be explained by two simple features: sentence length (SL) and sentence position within the passage (SP). Sentence length is captured by GPT2-XLU because the GELU nonlinearity in the first layer's MLP transforms normally distributed inputs with zero mean into outputs with a non-zero mean. This introduces a non-zero mean component to each token's representation in the residual stream. When these representations are sum-pooled, this non-zero mean component accumulates in a way that reflects the sentence length, making the length decodable in the intermediate layers (see A.9 for a formal proof). Sentence position is encoded within GPT2-XLU due to absolute positional embeddings which, although untrained, still result in sentences at the same position having similar representations when tokens are sum-pooled. We represent sentence position as a 4-dimensional one-hot vector, where each element corresponds to a given position within a passage, and sentence length as the number of words in a passage.

To obtain representations from GPT2-XLU, we selected the best-performing layer for each of the 10 untrained seeds. For EXP1 the best performing layer was layer 0 for 6 seeds, layer 1 for 3 seeds (first intermediate layer), and layer 19 for one seed. For EXP2 the best layer was layer 1 for 5 seeds, layer 2 for 4 seeds, and layer 5 for 1 seed.

We fit a regression using all subsets of the following feature spaces, SL, SP, GPT2-XLU, resulting in 7 models. For both experiments, $R^2_{SP+SL}$ was descriptively higher than all other models, including the best-performing model with GPT2-XLU (SP+SL+GPT2-XLU) (Fig. 2a). Sentence position was particularly important in EXP1, and sentence length was particularly important in EXP2. This may explain why the static layer often outperformed intermediate layer representations in EXP1 despite encoding sentence length more poorly. Overall, these results suggest that, when averaging across voxels within the language network in this dataset, GPT2-XLU does not improve neural encoding performance over sentence length and position.

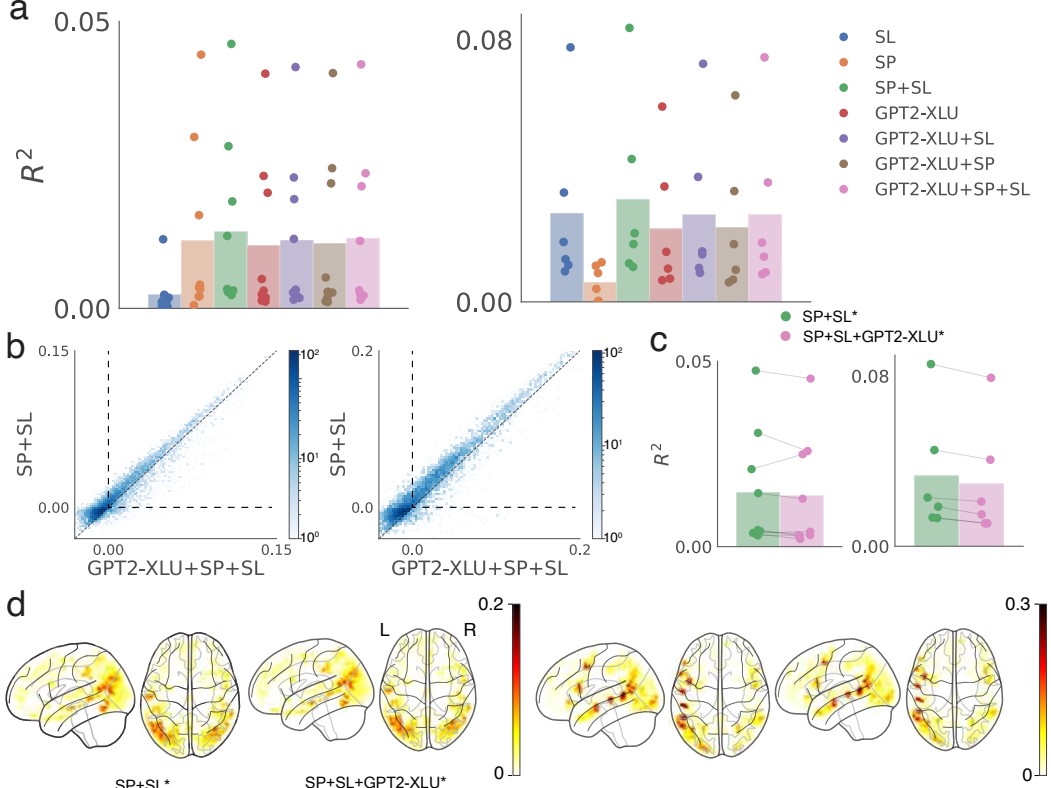

Figure 2: For all panels, EXP1 results are on the left and EXP2 results are on the right. **(a)** Brain score ($R^2$) for different combinations of features. Each dot represents $R^2$ values averaged across voxels in a single participant, with bars showing mean across participants. **(b)** 2D histogram of $R^2$ values for the best model without GPT2-XLU (SP+SL), and the best model with GPT2-XLU (GPT2-XLU+SP+SL). The dotted lines show $y = x$, $y = 0$, and $x = 0$. Values below $y = 0$ or left of $x = 0$ were clipped when averaging, but are shown here to visualize the full distribution. **(c)** Same as **(a)**, but after voxel-wise correction; lines connect data-points from the same participant. **(d)** Glass brain plots showing $R^2$ values of SP+SL (left) and GPT2-XLU+SP+SL (right) after voxel-wise correction. Conventions are the same as Figure 1.

Although GPT2-XLU did not enhance encoding performance when averaging across voxels, there may be a subset of voxels where GPT2-XLU does explain significant additional neural variance. To examine this possibility, we plotted a 2D histogram of voxel-wise $R^2_{SP+SL}$ values vs. $R^2_{SP+SL+GPT2-XLU}$ values in the language network (Fig. 2b). Values were clustered around the identity line, and there was no cluster of voxels where $R^2_{SP+SL+GPT2-XLU}$ appeared significantly higher. Next, for each voxel, we performed a one-sided paired $t$-test between the squared error values obtained over sentences (EXP1: $N = 384$ , EXP2: $N = 243$) between SP+SL+GPT-XLU and SP+SL. Across all functional networks, only 1.26% (EXP1) and 1.42% (EXP2) of voxels were significantly ($\alpha = 0.05$) better explained by the GPT2-XLU model before false discovery rate (FDR) correction; these numbers dropped to 0.001% (EXP1) and 0.078% (EXP2) after performing FDR correction within each participant and network [31]. None of the significant voxels after FDR correction were inside the language network. Taken together, these results suggest GPT2-XLU does not enhance neural prediction performance over sentence length and position even at the voxel level.

To control for voxels where the neural encoding performance of GPT2-XLU is weakened by the addition of SP+SL, we compared SP+SL* and SP+SL+GPT2-XLU*. When averaging across voxels, $R^2_{SP+SL}$* still exceeded $R^2_{GPT2-XLU+SP+SL}$* (Fig. 2c). Furthermore, the values for $R^2_{SP+SL}$* and $R^2_{GPT2-XLU+SP+SL}$* across brain areas were highly similar in both experiments (Fig. 2d). Only 1.00% (EXP1) and 1.18% (EXP2) of voxels were significantly better explained by the addition of GPT2-XLU before FDR correction; 0% (EXP1) and 0.05% (EXP2) of voxels were better explained

Table 1: Mean $R^2$ values (across participants) for each model. For models composed of multiple features, the best sub-model is used which includes the last feature.

| Features | EXP1 | EXP2 |
|---|---|---|
| GPT2-XL | 0.032 | 0.036 |
| SP+SL | 0.013 | 0.031 |
| SP+SL+WORD | 0.024 | 0.039 |
| SP+SL+WORD+SENSE | 0.026 | 0.040 |
| SP+SL+WORD+SENSE+SYNT | 0.027 | 0.043 |
| SP+SL+WORD+SENSE+SYNT+GPT2-XL | 0.032 | 0.045 |

after FDR correction (once again, no significant voxels were inside the language network ). Thus, our results hold even when controlling for decreases in performance due to the addition of feature spaces.

### 3.3 Sentence length, sentence position, and static word embeddings account for the majority of trained LLM encoding performance

We next turned to explaining the neural predictivity of the trained GPT2-XL. In addition to sentence position and sentence length, we added static word embeddings (WORD). Together, these features defined a baseline model which does not account for any form of linguistic processing of words in context. We next included three more complex features which involved contextual processing. First, we added sense-specific word embeddings from RoBERTa-Large using the LMMS package [32]. Sense embeddings contain distinct representations for different senses of the same word (e.g., mouse: *computer device*, and mouse: *rodent*). LMMS generates sense embeddings by averaging over contextual embeddings corresponding to the same sense of a word (see A.10 for further details).

Whereas sense embeddings help disambiguate many content words, they do not disambiguate pronouns, i.e., do not encode the entities that they refer to. Therefore, our sense embeddings were generated for a version of the Pereira text where pronouns were dereferenced (i.e., replaced by the words that they referred to). To maintain consistency with these sense embeddings, our static word embeddings were created (1) by taking a frequency-weighted average of sense embeddings for the same word, where frequency values were obtained from WordNet [33]; and (2) based on the dereferenced Pereira texts. Importantly, this means the impact of pronoun dereferencing and word and sense embeddings are not decoupled in this study. Finally, we created an abstract representation of the syntax of each sentence (SYNT), using an approach highly similar to that of Caucheteux et al. [34]: we collected sentences that are syntactically equivalent but semantically dissimilar to the original sentence, and averaged their representations from the best layer of GPT2-XL (A.11). We selected the best layer based on averaged $R^2$ across language voxels on test data (EXP1: layer 21, EXP2: layer 16).

We fit a regression to the fMRI data using all subsets of the feature spaces SL+SP, WORD, SENSE, SYNT, GPT2-XL, resulting in 64 models. In this list, features are ranked from least to most complex. For each feature, we took the model that exhibited the best performance in the language network which included that feature but did not include features more complex than it. For instance, values reported for $R^2_{SL+SP+WORD+SENSE}$ were taken from the best model which included SENSE, excluding models which included SYNT and GPT2-XL. By doing so, we were able to examine the impact of adding more complex features in explaining $R2_{GPT2-XL}$ while still accounting for decreases in test performance due to adding redundant features. We note that since this procedure is not performed at the voxel-level, we do not add a * to the $R^2$ notation.

Table 1 displays the performance of each model, including GPT2-XL on its own (Fig. 2a, 2b). The baseline SP+SL+WORD model, which does not account for any form of contextual processing, performs 75% as well as GPT2-XL in EXP1, and outperforms GPT2-XL in EXP2. When adding contextual features, namely SENSE and SYNT, our model performs 84.4% as well as GPT2-XL and the full model in EXP1, and better than GPT2-XL and 95.5% as well as the full model in EXP2, indicating that SENSE and SYNT play a modest role in accounting for GPT2-XL brain scores beyond simple features in this dataset.

Similar to previous sections, we perform voxel-wise correction by selecting the best sub-model with GPT2-XL and the best sub-model without GPT2-XL for each voxel. We focus only on sentence

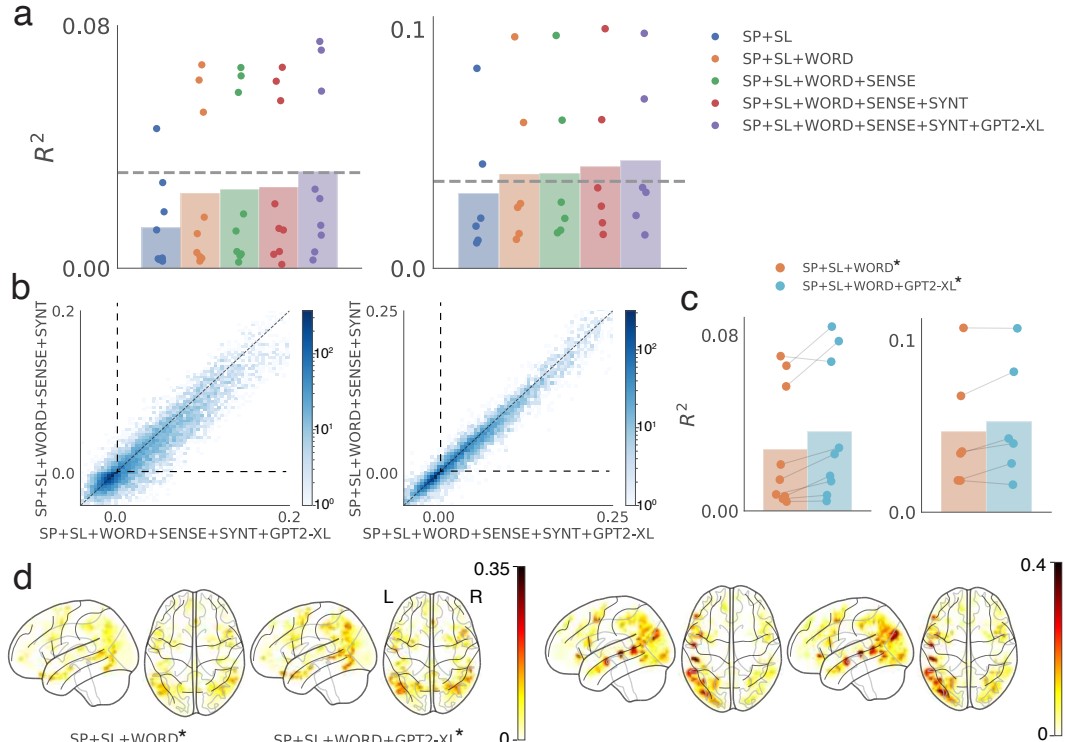

Figure 3: For all panels, EXP1 results are on the left and EXP2 results are on the right. **(a)** For each model, we display the sub-model which includes the added feature. Dots represent participants and bars are mean across participants. Grey dashed line is the performance of GPT2-XL alone. **(b)** 2d histogram comparing full model and full model with GPT2-XL. **(c)** Same as **(a)** but after voxel-wise correction for SP+SL+WORD and SP+SL+WORD+GPT2-XL. **(d)** Glass brain plots showing $R^2$ values of SP+SL+WORD (left) and SP+SL+WORD+GPT2-XLU (right) after voxel-wise correction.

position, sentence length, and static word embeddings because sense and syntax had modest contributions beyond these features. $R^2_{SP+SL+WORD}$* was 0.028 in EXP1 and 0.048 in EXP2, and $R^2_{SP+SL+WORD+GPT2-XL}$* was 0.036 in EXP1 and 0.056 in EXP2 (mean across participants) (Fig. 3c). This indicates that even after controlling for a reduction in GPT2-XL performance from the addition of simple features, GPT2-XL only explains an additional 28.57% (EXP1) and 16.7% (EXP2) neural variance over a model composed of features that are all non-contextual.

## 4 Fedorenko dataset

### 4.1 Shuffled train-test splits also impact ECoG datasets, but less than with fMRI

We first evaluated the impact of shuffled train-test splits on the Fedorenko dataset. Unlike in Pereira, the across-layer performance is well correlated between shuffled and contiguous splits ($r = 0.622$) (Fig. 4a). The OASM model performs 93.1% as well as GPT2-XL when averaging $R^2$ values across participants (Fig. 4b). $R^2_{OASM+GPT2-XL}$* was 45.3% better than OASM, meaning that the unique contribution of GPT2-XL is less than half the total contribution of a simple, auto-correlated model. Therefore, shuffled train-test splits also impact results on Fedorenko, albeit less than Pereira. This may be due to lower autocorrelation of ECoG compared to fMRI. We use contiguous splits for the remainder of the Fedorenko analyses.

### 4.2 Word position explains all of untrained, and most of trained, GPT2-XL brain score

As noted in [35], there was a strong positional signal in the ECoG dataset during comprehension of sentences that is likely related to the construction of sentence meaning. We therefore hypothesized

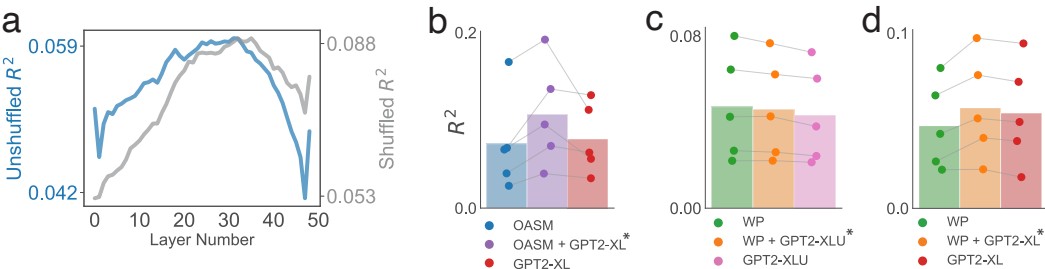

Figure 4: **(a)** Across-layer $R^2$, averaged across electrodes in the Fedorenko dataset, for GPT2-XL with and without shuffled splits. **(b)** Each dot is a participant, lines connect data-points from the same participant. Bars display mean across participants. **(c)** and **(d)** Same guidelines as **(b)**.

that a feature space that accounted for word position (WP) would do well relative to untrained and trained GPT2-XL. We generated a simple feature space that encodes word position, such that words in nearby positions were given similar representations (A.12). When performing a one-sided paired $t$-test between the squared error predictions of WP+GPT2-XLU* and WP, three electrodes were significantly better explained by the addition of GPT2-XLU before FDR correction, and none were better explained after FDR correction within each participant. Moreover, WP performs 86.7% as well as GPT2-XL, and 82.1% as well as WP+GPT2-XL*. Our results therefore suggest that the mapping between GPT2-XL and neural activity on the Fedorenko dataset is largely driven by positional signals.

## 5   Blank dataset is predicted at near chance levels

Lastly, we address the Blank dataset. We find that OASM achieves an $R^2$ that is 103.6 times larger than that of GPT2-XL when using shuffled splits A.13, demonstrating that such splits are massively contaminated by temporal autocorrelation. We next turn to using contiguous splits, and test whether GPT2-XL performs better than an intercept only model by applying a one-sided paired $t$-test between the squared error values obtained from GPT2-XL and the intercept only model ($N = 1317$ TRs). GPT2-XL predicts 1 fROI significantly better than an intercept only model, and 0 fROIs are significantly better after FDR correction. Our results therefore suggest that GPT2-XL performs at near chance levels on the version of the Blank dataset used by [2, 10, 11].

## 6   Limitations and Conclusions

Our study has three main limitations. First, our method of examining how much neural variance an LLM predicts over simple features scales poorly when the number of features is large. Second, although we attempted to correct for cases where adding features decreases test set performance and employed banded regression, fitting regressions with large feature spaces on noisy neural data with low sample sizes can lead to poor estimations of the neural variance explained. Finally, we did not analyze datasets with large amounts of neural data per participant, for instance [36], in which the gap between the neural predictivity of simple and complex features might be much larger.

In summary, we find that on the Pereira dataset, shuffled splits are heavily impacted by temporal autocorrelation, untrained GPT2-XL brain score is explained by sentence length and position, and trained GPT2-XL brain score is largely explained by non-contextual features. We find that the majority of GPT2-XL brain score on the Fedorenko dataset is accounted for by word position, and on the Blank dataset GPT2-XL predicts neural activity at near chance levels. These results suggest that (i) brain scores on these datasets should be interpreted with caution; and (ii) more generally, analyses using brain scores should be accompanied by a systematic deconstruction of neural encoding performance, and an evaluation against simple and theoretically uninteresting features. Only after such deconstruction can we be somewhat confident that the neural predictivity of LLMs reflects core aspects of human linguistic processing.

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

# A Appendix

## A.1 Experimental data

**Pereira:** For both experiments, each sentence was visually presented for 4 s with 4 s between sentences and an additional 4 s between passages. A single fMRI scan was taken in the interval between each sentence. Because fMRI data is noisy, each experiment was repeated three times and fMRI data was averaged across the repetitions. A single fMRI scanning session consisted of 8 runs, where each run contained 12 passages in EXP1 and 9 passages in EXP2. Participants performed a total of 3 scanning sessions. The division of passages into runs and the order of the runs was randomized for each participant and scanning session.

**Fedorenko:** Participants read sentence on word at a time, and each word was visually displayed for 450 or 700 ms. For each electrode, high gamma signal was extracted using gaussian filter banks at center frequencies ranging from $73 - 144$ Hz, the envelope of the high gamma signal was computed through a hilbert-transform, and the envelope was z-scored within each electrode. For each participant, language-selective electrodes were selected where the z-scored envelope of the gamma activity was significantly higher during the sentences than a condition where participants read nonword lists. Z-scored high gamma activity from these language-selective electrodes were used in subsuquent analyses.

**Blank:** Text was split into 2 s segments corresponding to each TR, with words that were on the boundary being assinged to the later TR. Due to the delay in the hemodynamic response function (HRF), neural activity was predicted using stimuli from 2 TRs (4 s) previous.

**Functional localization:** For Pereira and Blank, the language network was defined by the following procedure [19]. First, voxels were identified in each participant which showed stronger responses to sentences compared to lists of non-words (sentences > non-word lists contrast). These voxels were then constrained by data-driven language activation maps formed by applying the same contrast to many other participants. Finally, the top $10\%$ of the voxels were selected which showed the greatest sentences > non-word lists difference. For Pereira, we perform some analyses using four other networks: multiple demand (MD), default mode network (DMN), auditory, and visual network. The multiple demand (MD) and default mode network (DMN) networks were defined using the same procedure, except that the contrast involved a spatial working memory task, where a hard > easy condition contrast was used for MD and a fixation > hard contrast was used for DMN [37]. Auditory and visual networks were defined using resting state connectivity [38].

## A.2 Banded ridge regression

We used a random search method to optimize the banded regression hyperparameters [28]. Banded regression has two hyperparameters, $\gamma$, which is a vector of shape number of feature spaces that determines how much each feature space is scaled, and $\alpha$, which is the L2 penalty applied across feature spaces. Values for $\gamma$ are drawn from a Dirichlet distribution and hence sum to 1. Down-scaling a certain feature space relative to others is functionally equivalent to assigning a separate L2 penalty for each feature space. This is because when a feature space is down-scaled, the L2 magnitude of the weights must increase for it to have a meaningful contribution to the predictions, which equates to increasing the L2 penalty for that feature space. The optimal $\gamma$ and $\alpha$ combination was found for each voxel/electrode/fROI by performing a random search over $\gamma$ values, storing the $\alpha$ value that performed best for that $\gamma$ on validation data, and then selecting the best performing $\gamma$ and $\alpha$ combination.

Before starting the random search, we tried all combinations of $\gamma$ values that removed feature spaces (i.e. down-scaled at least one feature space to 0) to ensure the regression had an opportunity to remove features which hurt performance. In theory, this should obviate the need for the procedure implemented in 2.8. This is because the banded regression procedure can remove feature spaces based on validation data, meaning if a model performs worse than a sub-model the banded procedure has the opportunity to set the $\gamma$ value corresponding to the additional feature spaces to 0. However, because neural data is noisy and there is often little data per subject, performance on validation data is not always indicative of performance on test-data. Therefore it is possible for the banded regression procedure to include a feature space (since it helps on validation data), and for this feature space to ultimately hurt test set performance, necessitating the correction procedure detailed in 2.8.

We ran banded ridge regression for a maximum of 1000 random search iterations with early stopping if the mean $R^2$ did not improve by more than $10^{-4}$ after 50 iterations. We treated feature spaces with many dimensions as one features because preliminary results showed this performed better. Specifically, we always treated the following feature spaces as one feature space: static word embeddings, sense-specific word embeddings, syntactic representations, and GPT2-XL and Roberta-Large representations. All other features were treated as their own feature space.

We z-score all features across samples before training regressions, as is standard when using ridge regression in neural encoding studies.

## A.3 Additional details on train, validation, and test folds

**Pereira:** During each outer fold, a single passage from each of the 24 semantic categories from one experiment was selected, and half of these passages were designated as the test set. This equated to 8 test folds for experiment 1 (4 passages per semantic category) and 6 test folds for experiment 2 (3 passages per semantic category). During each inner fold, we again selected one passage from each semantic category, and half of these passages were designated as validation (leading to 7 inner folds for experiment 1, and 5 inner folds for experiment 2).

528 **Fedorenko:** For each outer fold, we selected 4 sentences as the test fold, resulting in 13 outer folds.
529 For each inner fold, we once again select 4 sentences as the validation set, resulting in 12 inner folds
530 per outer fold.

531 **Blank:** For each outer fold, we selected a single story as the test fold, resulting in 8 outer folds. For
532 each inner fold, each of the remaining stories served in turn as the validation set, resulting in 7 inner
533 folds.

### A.4  Justification of statistical tests

535 We performed a *t*-test between squared error values from two models to determine if one model
536 performs better than another. While squared error values are not always normally distributed, our
537 sample sizes were large (the minimum sample size was 243) and so we still opted to use a t-test over
538 a non-parametric alternative [39]. One issue with a t-test is that relies on the assumption that samples
539 are not correlated, which is not true for time-series data. However, we note that correlated samples
540 leads one to underestimate the standard error of the mean and exaggerate differences between two
541 models. Since we only perform one-sided t-tests to examine whether adding GPT2-XL representations
542 improves performance, the net impact of this on our results is to overestimate how much GPT2-XL
543 contributes over simple features.

### A.5  Across layer $R^2$ values in the Pereira dataset

545 Across layer performances in the Pereira dataset for GPT2-XLU and GPT2-XL when using the sum
546 pooling method (Fig. 5a,b) and the last token method (Fig. 5c,d). Performance in language network
547 is higher across the board than performance in DMN, MD, and visual networks. We do not show
548 auditory network results because participants read passages in Pereira and hence auditory brain scores
549 are near 0. Furthermore, performance is lower with the last token method in every case except in
EXP1 trained results where the last token method performs slightly better.

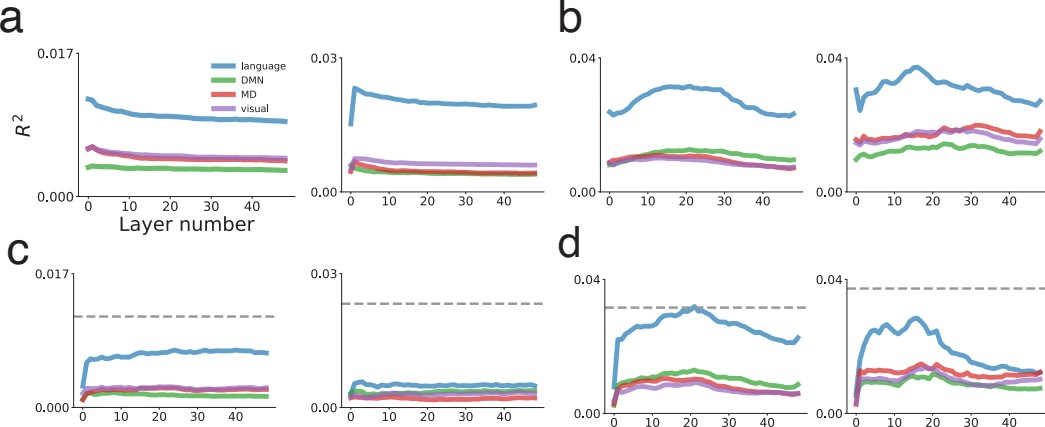

Figure 5: **a)** Across layer performances in Pereira dataset for GPT2-XLU for each functional network
when using the sum-pooling method. EXP1 is on the left, and EXP2 is on the right. **b)** Same as **a** but
for GPT2-XL, also using the sum-pooling method. **c)** Same as **a** but when using the last token method.
Dotted grey line shows performance of best layer of GPT2-XLU in language network when sum
pooling. **d)** Same as **b** but when using the last token method. Dotted grey line shows performance of
best layer of GPT2-XL in language network when sum pooling.

550

### A.6  RoBERTa-Large shows similar results as GPT2-XL

552 To examine whether our results depending on the choice of LLM, we replicated all of our Pereira
553 trained analyses with RoBERTa-Large (ROB). The overall trend in results was the same as
554 with GPT2-XL (Fig. 6). Namely, SP+SL+WORD performed 76.8% as well as the full model
555 (SP+SL+WORD+SENSE+SYNT+ROB) and 80.0% as well as ROB alone in EXP1, and in EXP2 it

performed 88.0% as well as the full model and better than ROB. Furthermore, SENSE and SYNT bridge the gap to the full model by a small amount. In sum, our main conclusion that a large amount of trained LLM brain score in the Pereira dataset is accounted for by non-contextual features also applies to RoBERTa-Large.

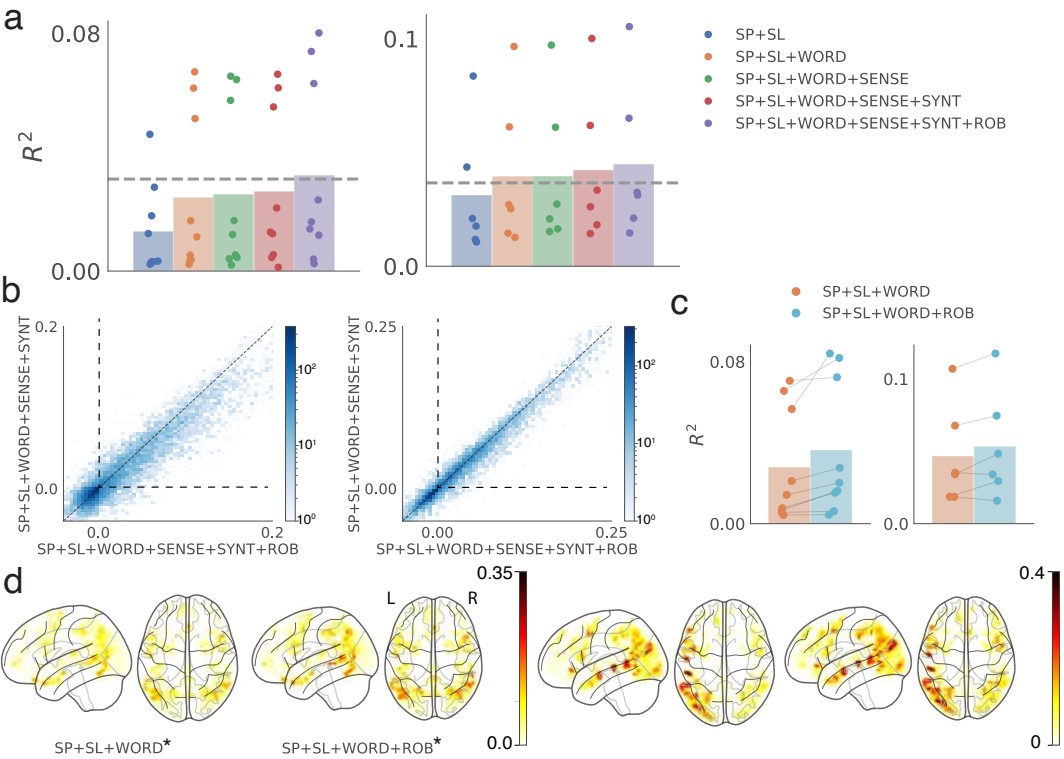

Figure 6: All panels are the same as Figure 3, except GPT2-XL is replaced with RoBERTa-Large (ROB).

## A.7 Orthogonal autocorrelated sequences model (OASM) hyperparameters

The width of the Gaussian filter used for within-block smoothing was $\sigma = 2.2$ in Pereira, $\sigma = 1.8$ in Fedorenko, and $\sigma = 1.5$ in Blank. Gaussian widths were determined by sweeping $\sigma$ across 50 evenly spaced values between 0.1 and 5.0 and choosing the best-performing $\sigma$ for each dataset.

## A.8 Shuffled train test splits confound task-relevant and task-irrelevant neural activity

OASM is a model which clearly lacks any linguistic representations that would allow it generalize to fully held-out passages. However, this is is not to say that OASM is not correlated with linguistic features. For instance, sentences in a given passage are more semantically related with each other than with sentences in other passages. Nonetheless, using shuffled train-test splits almost certainly exaggerates the variance explained by a model which, on the basis of semantic similarity, arrives at a similar representational structure as OASM. This is because task-irrelevant neural responses make up a large fraction of neural activity [40], and shuffled train-test splits allow a model with OASM-like representational structure to predict not just the task-relevant neural responses driven by the participant reading the passage, but also any task-irrelevant neural activity that was present throughout the reading of the passage. Hence, we strongly urge researchers to avoid shuffled train test splits when evaluating the neural predictivity of language models, and we surmise that previous studies using shuffled train-test splits to compare neural predictivity between models might have come to erroneous conclusions.

## A.9 Linear decodability of sentence length

Here, we show that the MLP block adds a linearly decodable component with non-zero mean to the residual stream in the GPT2 architecture.

**Proof** :

We denote the $i$'th input to the MLP block in the first layer of GPT2-XL as $x_i$. The output of the MLP block is defined as follows:

$$MLP(x_i) = x_i + W_d(GELU(W_u(LayerNorm(x_i))))$$

We assume that the elements of $x_i$ are normally distributed. For a given $x_i$, it then follows that the distribution of elements in $LayerNorm(x_i)$ is normal with $\mu = 0$ and $\sigma = 1$ (assuming the standard $LayerNorm$ initialization).

Because $W_u$ is initialized from a zero-mean normal distribution, $W_u(LayerNorm(x_i))$ also has zero-mean.

Note that $GELU$ is a function for which $\mathbb{E}[Y] > 0$ for $Y$ normally distributed with mean 0. Hence, the mean value across elements following the $GELU$ is non-zero. Let us denote this mean value across all elements of $GELU(W_u(LayerNorm(x)))$ and across all tokens $x$ as $m$. Then, for an MLP with up-projected dimension $d_u$, we can take the dot product of $GELU(W_u(LayerNorm(x_i)))$ and $\frac{1}{d_u m} \times \hat{k}$, where $\hat{k}$ is a $d_u$-dimensional vector of 1s. The resulting value will have mean 1.

However, we cannot decode this value directly from the MLP in practice; first, this vector is down-projected back to the residual stream by $W_d$. Nonetheless, we can still closely approximate it, assuming it is approximately orthogonal to $x_i$, by using the pseudo-inverse of $W_d$. More specifically, we can extract a scalar with mean 1 as follows:

$$\sqrt{\frac{d_u}{d_d}} \times \frac{1}{d_u m} \times \hat{k} W_d^\dagger MLP(x_i)$$

where $d_d$ is the down-projected dimension. Because this extracted scalar value is distributed with mean 1 across token representations $x_i$, assuming independence of token representations within a sentence, the sum of the extracted scalar value across the tokens of a sentence is distributed with mean equaling the number of tokens in the sentence.

## A.10 LMMS

LMMS generates a sense embedding for each word by averaging across contextual embeddings (in our case from RoBERTa-Large) of that sense derived from a sense-annotated corpus. For words in WordNet where labeled senses don't exist, LMMS sets their sense embeddings equal to the average of sense embeddings with the same sense (or same hypernym/lexname if that approach fails). Finally, the sense embeddings are averaged together with the gloss embeddings for that sense of the word generated using the same LLM. For additional details refer to Loureiro et al. [32].

## A.11 Contextual syntactic representations

Syntactic embeddings are derived by substituting content words (nouns, verbs, adjectives, and adverbs) in the original sentences with words from the Generics KB corpus, matching their part-of-speech and dependency tag via the SpaCy transformer-based tagger [41]. For each sentence in the Pereira dataset, we generate 170 new sentences, ensuring the subtree token indices from each token match those of the original sentence. The top 100 sentences, selected based on summed surprisal with GPT2-XL, are retained. Each sentence's syntactic embedding is then computed by summing token representations within each sentence and then averaging across the 100 sentences.

## A.12 Word position feature in Fedorenko dataset

The primary finding in the paper which first collected the Fedorenko dataset [35] was a ramping of neural activity across the words of sentences, where each sentence was 8 words long. Hence, we concatenate a linearly ramping 1-dimensional positional signal to an 8-dimensional 1-hot positonal

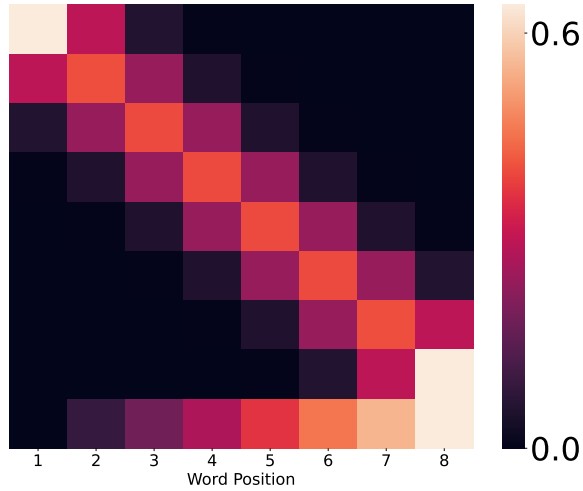

Figure 7: Word Position feature for a single sentence in the Fedorenko dataset.

signal. Because we expect positional signals to be more simlar between adjacent words than more
distant words, we apply a Gaussian filter ($\sigma = 1$) to the 8-dimensional positional signal. The resulting
feature space, which we refer to as "word position" in the main text, is shown for a single sentence in
the above figure.

### A.13 OASM and GPT2 Model Comparison on Blank Dataset

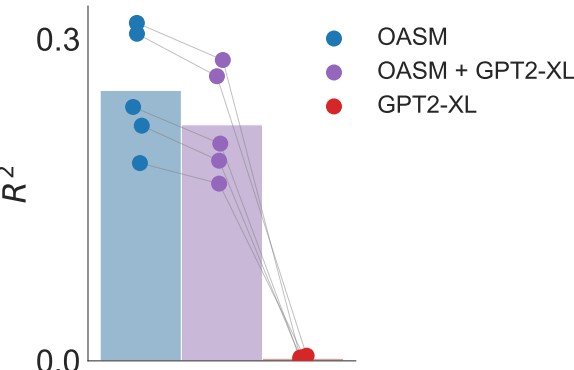

Figure 8: OASM far outperforms GPT2-XL on the Blank dataset, and GPT2-XL does not appear to
explain any variance beyond that explained by OASM.

We find that OASM achieves 103.6 times higher neural predictivity than GPT2-XL on the Blank
dataset when using shuffled train-test splits. There could be several reasons for this. First, it might
be that the method for pooling representations from GPT2-XL used here 2.3 and in [2, 10, 11]
did not yield useful enough representations for GPT2-XL to map effectively to the brain data. An
additional likely culprit is that, of the three datasets we study here, Blank has the greatest potential for
autocorrelation in temporally adjacent samples. This is because, while the Pereira dataset typically
has a TR every 8 seconds, the Blank dataset has a TR every 2 seconds. We note that our results here
are not completely surprising; given that [2, 10] observed untrained GPT2 models perform far better
than trained models on this dataset, it did not seem likely that GPT2-XL would map onto neural
representations of linguistic features here.

### A.14   Computational Resources

All analyses were done between 2 machines: One with 2 RTX 3090 GPUs, and another with 1 RTX 4090 GPU. The most computationally demanding parts of our analyses were fitting the banded ridge regressions used to generate Figure 3, collecting untrained model results across 10 seeds, and generating syntactic representations, which each took around 3 hours to complete.

### A.15   Dataset Licenses

The Blank dataset was originally released as part of the Natural Stories Corpus, which is provided under the CC BY-NC-SA license [23]. The Pereira dataset is released under the Creative Commons License [8]. The version of the Fedorenko dataset used here is provided under the MIT license. All datasets used are the same versions as in [2] and can be downloaded using the neural-nlp repository: `https://github.com/mschrimpf/neural-nlp/tree/master`. All datasets were collected with IRB approval at their respective institutions.

