# OpenReview forum: "What Are Large Language Models Mapping to in the Brain? A Case Against Over-Reliance on Brain Scores"
_NeurIPS.cc/2024/Conference — Submitted to NeurIPS 2024_

### Official Review · Reviewer_bjuX · 2024-06-25

**Soundness:** 3
**Presentation:** 2
**Contribution:** 3
**Rating:** 7
**Confidence:** 3

**Summary:**

The paper studies the existing "brain score" approach of evaluating how similar LLM representations are to human brain activity. First, they show that when using shuffled train-test splits on the Pereira dataset, which some prior studies use, a trivial temporal auto-correlation model performs similarly to GPT2-XL. Second, they show that untrained GPT2-XL's brain score is simply due to encoding sentence length and position. Third, they show that a trained GPT2-XL's brain score is largely explained by sentence length, sentence position, and static word embeddings, which are all non-contextual features.

**Strengths:**

1. The paper studies the important topic of understanding why recent research has shown similarities in LLM representations and human brain language activity.
2. They highlight issues with existing neural datasets commonly used in the LLM-brain field, e.g., shuffled train-test splits on the Pereira dataset.

**Weaknesses:**

From most to least significant:
1. The paper writes: "OASM out-performed GPT2-XL on both EXP1 and EXP2 (Fig. 1b, blue and red bars), revealing that a completely non-linguistic feature space can achieve absurdly high brain scores in the context of shuffled splits. This strongly challenges the assumption of multiple previous studies [2, 11, 10] that performance on this benchmark is an indication of a model’s brain-likeness" (Lines 173-177).
- I agree this shows that a model that exploits temporal auto-correlation, OASM, can achieve similar neural predictivity on the Pereira dataset as GPT2-XL. However, this does not necessarily mean that GPT2-XL's neural predictivity is attributed to temporal auto-correlation, rather than linguistic similarity. It also does not tell us the proportion of GPT2-XL's neural predictivity that can be attributed to each factor. Although GPT2-XL can theoretically exploit temporal auto-correlation artifacts, it may not be empirically doing so as it was optimized for language performance instead.
- Furthermore, OASM may be a much stronger method at exploiting temporal auto-correlation than GPT2-XL's architecture is capable of. The paper's results may highlight that the Pereira dataset is easy to "cheat" using temporal auto-correlation, but not that GPT2-XL or other LLMs are doing so.
2. The paper evaluates "brain score" using a metric they defined, out-of-sample R-squared, whereas the prior research they cite [2, 24] seemed to use Pearson correlation. Although they argue for the advantage of the metric they used, it is challenging to understand how their results relate to prior research. For example, they do not show the Pearson correlation that their OASM model obtains on Pereira, which would make it easier to compare to models in prior research. Furthermore, they only tested a single language model, GPT2-XL, whereas more recent research has used larger or different models.
- Additionally, the metric they defined seems to produce results close to 0 for GPT2-XL and less than 0.05 for all models too. In Figure 2b, the R-squared results cluster around zero, with many negative values. They obtain an average R-squared value that is positive (e.g., Figure 2a?) only because they clip negative values when averaging.
3. The paper provides a theoretical justification arguing that GPT2-XL can encode sentence length and sentence position (Lines 197-201). However, this does not necessarily mean that GPT2-XL's neural predictivity is attributed to sentence length/position, rather than contextual/semantic features. It also does not tell us the proportion of GPT2-XL's neural predictivity that can be attributed to the two factors.
- They compared GPT2-XL to two ideal models of sentence position (SP, represented as a 4-dimensional one-hot vector) and sentence length (SL, represented as a scalar). However, these ideal models may be a much "cleaner" representation of sentence length/position than the perhaps noisy GPT2-XL representation of sentence length/position that may not be cleanly and linearly decodable.
4. The paper writes: "GPT2-XL only explains an additional 28.57\% (EXP1) and 16.7\% (EXP2) neural variance over a model composed of features that are all non-contextual." However, the paper does not provide a noise ceiling for the metric they defined, out-of-sample R-squared. Consequently, it is unclear whether the small improvements in neural predictivity is due to hitting the noise ceiling.

**Questions:**

Just for clarification:
1. Why is the R-squared brain score of the same model, GPT2-XL trained, so much higher in Figure 1b (R-squared of ~0.12) compared to Figure 1a (R-squared of ~0.03) and Table 1 (R-squared of ~0.035)? Is there some difference in the models in each figure that I have misunderstood?

**Limitations:**

Limitations not mentioned in the paper:
1. Please see Weaknesses 1-4.

---

> ### Author Rebuttal · Authors · 2024-08-07
>
> **W1**: Thank you for raising this point. Indeed, the percentages we reported in the submitted draft suggest that the relative increase in variance explained by OASM+GPT2-XL above OASM alone is fairly small (13.6% (EXP1) and 31.5% (EXP2)). While this does explicitly show that the variance that GPT2-XL uniquely predicts beyond what is predicted by OASM is small compared to what OASM predicts alone, indicating that brain scores on these datasets are more sensitive to a model’s autocorrelation pattern than to its linguistic features, you are correct that it does not explicitly state what proportion of the variance predicted by GPT2-XL can also be explained by OASM. To address this, we’ve added new variance partitioning metric, termed $\Omega$, which we explain in detail in the global response section.
>
>  $\Omega_{\textit{GPT2-XL}}(\textit{OASM})=$ $81.5 \pm 5.5$ in EXP1, and $62.7 \pm 4.7$ in EXP2, meaning that it is the case that a large majority of what GPT2-XL is explaining in *Pereira* can be accounted for by OASM, and hence can be attributed to it effectively capturing autocorrelation. This is also the case in *Fedorenko*, where $\Omega_{\textit{GPT2-XL}}(\textit{OASM})=$ $56.8 \pm 4.9$, as well as in Blank, where $\Omega_{\textit{GPT2-XL}}(\textit{OASM})=$ $100 \pm 0.0$ (since every fROI is predicted worse by OASM+GPT2-XL than by OASM alone). We will include these analyses in the manuscript to show GPT2-XL predictivity can largely be explained by autocorrelation when using shuffled train-test splits.
>
> **W2**: We provide results using Pearson correlation and a battery of other design choices, including the particular combination used in Schrimpf et al. 2021, in the rebuttal pdf, and we will incorporate these figures into the Appendix of our paper. As we explain in our global response, we wanted a metric that could be interpreted as the fraction of variance explained, something that R2 affords but Pearson correlation does not. As shown in Figure 2 of our supplement, when using Pearson correlation and all of the other design choices of Schrimpf et al. 2021 (linear regression, last token, median across voxels within each participant), OASM continues to outperform GPT2-XL.
>
> In regards to the language model used, we also performed our analyses with RoBERTa-Large in Appendix A.6 and saw the same key results. We chose to focus on GPT2-XL because Schrimpf et al. 2021 identified it as the highest performing model across these 3 datasets. Also, running our analyses with larger models would require considerable changes to our code to avoid out-of-memory errors, which we did not have time to address during the rebuttal period. We can reproduce our main analyses with 7B and 13B Llama 3 models for the camera-ready version if desired. We address the clipping concern in a comment below.
>
> **W3**: The theoretical justification in lines 197-201 is about the *untrained* GPT2-XL model, which we refer to as GPT2-XLU. Empirically, we show that GPT2-XLU does not explain any additional variance over sentence position (SP) and sentence length (SL) by incorporating GPT2-XLU, SP, and SL into a single banded ridge regression, and finding that it does not produce significantly better predictions in a single language network voxel (after within-participant FDR correction) than SP+SL. Indeed, we also find that GPT2-XLU is outperformed by SP+SL, and this might be due to GPT2-XLU containing noisier representations of SP and SL. However, if GPT2-XLU contained any representations that were helpful for predicting brain activity beyond SP and SL, we would expect to see the combined regression (SP+SL+GPT2-XLU) predict neural variance above SP+SL. That we don’t find increased neural predictivity in a single voxel is strong evidence that GPT2-XLU is not mapping to anything that can’t be predicted by SP and SL. Regarding the proportion of (trained) GPT2-XL’s neural predictivity that can be attributed to the SP+SL, we find that   $\Omega_{\textit{GPT2-XL}}(\textit{SP+SL})=$ $39.1 \pm 7.8$ in EXP1 and $60.7 \pm 10.2$ in EXP2, meaning that a sizable chunk of GPT2-XL brain score can be explained by SP+SL, particularly for EXP2.
>
> **W4**: This is an interesting point. If the small magnitude of the additional variance explained by GPT2-XL were due to hitting the noise ceiling, then non-contextual features are sufficient to explain almost all of the explainable variance in this dataset. If so, such simple features really are almost all you need to explain not only what GPT2-XL is mapping to in this dataset, but the dataset itself. We will add additional discussion in the text to address this.
>
> We emphasize our comparisons are relative, so that a noise ceiling cancels out when considering the ratio of variance explained by simple features vs GPT2-XL. We also note that noise ceilings, when calculated by predicting each participant’s voxels with all other participants’ voxels and then extrapolating the performance to infinitely many participants, typically underestimate how much variance can be explained in a dataset. For instance, in a more recent paper using the same analysis choices as Schrimpf et al. 2021, several LLMs exceed the “noise ceiling” in Pereira, often by a large margin (Aw et al. 2023). Lastly, computing the noise ceiling in the same style as in Schrimpf et al. 2021 is quite computationally expensive. For all of these reasons, we were unable to compute a noise ceiling prior to the end of the rebuttal period, but would be willing to include it in the camera ready version if desired.
>
> **Q1**: Figure 1b shows results when using shuffled train-test splits. Figure 1a shows results for both unshuffled (blue line, left y-ticks) and unshuffled (gray line, right y-ticks) train-test splits. As you’ve noted, the unshuffled train-test splits (R-squared ~0.03) yield considerably lower performance than the shuffled train-test splits (R-squared ~0.12). This is to be expected, and we will make this more clear in the figure caption.

---

> > ### Comment · Reviewer_bjuX · 2024-08-09
> >
> > I raised my score from 5 to 7. The authors provided convincing clarifications for most of the weaknesses (and their sub-points) I raised.
> >
> > 1. Whether GPT2-XL is exploiting temporal auto-correlation rather than linguistic similarity (mostly resolved)
> > - Thanks for the additional variance partitioning analyses.
> > 2. Use of new metrics and design choices that make it hard to contextualize to prior work (resolved)
> > - Thanks for showing new results using Pearson correlation and other design choices used in prior work
> > 3. Proportion of GPT2-XL's neural predictivity that can be attributed to SP and SL (resolved)
> > - Thanks for bringing up the result that the ridge regression model of GPT2-XLU+SP+SL does not produce significantly better predictions than GPT2-XLU.
> > - Thanks for the additional variance partitioning analyses for the trained GPT2-XL.
> > 4. Noise ceiling (resolved)
> > - Thanks for the clarifications.
> >
> > Additionally, I appreciated your additional explanation of shuffled train-test splits and why they are not great, in your rebuttal to Reviewer 5zfv.

---

> ### Author Response · Authors · 2024-08-07
> **Why we clip when averaging across voxels**
>
> In general when evaluating R2 on held-out data, values can range from $-\inf$ to $1$. If some voxels are predicted at worse than chance levels (yielding negative R2), this can drag down the mean R2 we report across voxels, even if some other voxels are actually predicted quite well. Although clipping would be problematic if our goal were to predict neural activity as well as possible, in which case it would inflate our reported performance, we believe that it is justified given that our goal is to explain the neural variance explained by LLMs in neural activity. If, for example, we found that adding an LLM onto a simple set of features brought down the mean R2, we might naively conclude that our job is done and that there is nothing that the LLM is explaining in addition to our simple features. However, it is very possible that the LLM is in fact explaining additional variance in some fraction of voxels, but overfitting on other voxels which it cannot actually predict, and that the negative R2 from the overfit voxels washes out the improved predictivity on the better-predicted voxels when averaging. Ultimately, we chose to clip R2 values at 0 when averaging as a conservative approach to prevent such cases from influencing our measure of variance explained.
>
> As for the many poorly predicted voxels, indeed, there was a large cluster of voxels with near-zero R2. Some voxels simply can't be predicted very well, even by GPT2-XL. We argue that this does not reflect a weakness of our methods, but rather a fact of the data.

---

> ### Author Response · Authors · 2024-08-09
>
> Thank you for your thoughtful response and for raising your score based on the clarifications we provided. We're glad to hear that our additional analyses and explanations addressed your concerns effectively.
>
> If there are any further questions or aspects you think could still benefit from more clarification, particularly regarding whether GPT2-XL is exploiting temporal auto-correlation rather than linguistic similarity, please let us know. We appreciate your feedback and the opportunity to improve our work.

---

### Official Review · Reviewer_VpMi · 2024-07-03

**Soundness:** 3
**Presentation:** 3
**Contribution:** 2
**Rating:** 5
**Confidence:** 5

**Summary:**

The authors investigate the simplest set of features that can explain variance in neural recordings (fMRI, ECoG) during language processing. The authors focus on the surprisingly high alignment ("brain scores") of untrained LLMs, but also investigate trained LLMs. The authors conclude that the predictivity performance of untrained LLMs can be explained by simple features such as sentence position and sentence length. The authors quantify the effect of autocorrelation on shuffled cross-validated train-test splits and find that predictors that account for the temporal structure in the neural data explain the data better than other (linguistic) features. Overall, the study highlights the importance of understanding why LLMs (or, any feature space for that sake) map onto the brain.

**Strengths:**

- The paper is generally well-written, and the analyses are well-motivated. The topic is timely.
- The paper is very comprehensive, and provides in-depth analyses of one widely used dataset (from Pereira et al. 2018) for LLM-brain mapping studies. The paper also investigates two other datasets, but in less depth. The authors run analyses across several seeds for the untrained models, and in general, include a good amount of well-motivated control analyses.
- The analyses of trained GPT2-XL are interesting (Section 3.3), and provide a good contrast to the analyses of the untrained models.

**Weaknesses:**

- The authors motivate the paper with "attempting to rigorously deconstruct the mapping between LLMs and brains", but do not really acknowledge other work doing so. The paper lacks a short relevant work section on other studies that ask why artificial models map onto human brain data (from language, e.g., Merlin and Toneva, 2022; Kauf et al. 2023; Gauther and Levy, 2019, ...).
- I find it very odd that the authors include "brain scores" in their title and also motivate the study through Schrimpf et al. 2021, but then do not replicate almost any of the analysis choices in Schrimpf et al. 2021: the feature space pooling is different, the ridge regression, the evaluation metric. For instance, sum feature pooling is motivated because "it provides higher alignment scores", but other studies motivate last token pooling because it is conceptually better motivated (Transformers integrate over the context). It does not feel quite right to make decisions based on "what gives the highest alignment", because, perhaps sum feature pooling does indeed artificially inflate scores. Either way, it is not very suitable to link the title and most of the motivation of the paper based on one instance of prior work, and then make completely different choices. That being said, the choices are definitely well-motivated in most cases, but it makes the comparison with prior work different -- which is fine, the motivation should just be changed in that case.
- The authors should make it more clear which voxels are used in which analyses. The authors mention that unless otherwise noted, the language voxels are used (line 75), but it is not always very clear. For instance, Figure 2d clearly includes voxels across several networks.
- Regarding novelty: Kauf et al. 2023 also investigated contiguous splits on Pereira2018 as a supplementary analysis (not to the same extent as in the current paper), and also discusses the problem of temporal auto-correlation.

**Questions:**

- I do not understand how the plots in Figure 2b and Figure 2d match up, and how those match with the text. From Figure 2d, it is evident that a lot of voxels are predicted around ~0 by both feature spaces. From Figure 2d, it looks like most voxels that are well predicted by SP+SL are close to visual regions? Which would make sense, because those regions should care more about the perception of characters, i.e., SL makes a lot of sense here. Can you include a discussion of that, please? Moreover, it seems to be the case that the voxels that are best predicted by SP+SL+GPT-XLU are around language regions in temporal and frontal lobes (left hemisphere), but that does not align with the statistics reported in the text? Also, there are some voxels below the diagonal in Figure 2b, where are those?
- I am missing a link between OASM and contextualization of LLMs. As seen in previous work, contextualization/positional encoding of LLMs can provide the "wrong" information in cross-validation (Antonello et al. 2023; Kauf et al. 2023). How does OASM relate to contextualization of LLMs? Isn't it two sides of the same coin? Any predictor that "bleeds over" information between cross-validation splits will give high scores for wrong reasons? So for instance, by not allowing LLMs to contextualize across splits, the problem is partly solved?
- I don't fully understand the motivation for the correction described in Section 2.8 -- in the authors' cross-validation scheme, why is it the case that adding *additional* predictors decrease the scores? Overfitting on the train set?

**Limitations:**

The authors discuss limitations of their study.

---

> ### Author Rebuttal · Authors · 2024-08-07
>
> **W1**: We agree, and will incorporate a related works section into our paper.
>
> **W2**: Thank you for pointing this out. We have replicated all our key findings in *Pereira* using the design choices made in Schrimpf et. al 2021, specifically we used last token pooling, computed Pearson correlation in the same manner as Schrimpf et. al 2021, and used vanilla linear regression (see Figure 1 and Figure 2 in pdf). We also replicated our key findings with other permutations of these design choices, for instance using banded ridge regression, last token pooling, and Pearson correlation (Figure 1) to ensure our study is robust to a wider range of design choices. We have provided more detail on this, as well as an expanded discussion on why we use banded ridge regression, out of sample R2, and sum pooling in the global rebuttal.
>
> **W3**: Yes that is completely correct, Figure 2d and 3d show results for all functional networks in the Pereira dataset (visual, auditory, multiple demand, default mode network, and language), and we will clarify this in our updated draft. We will make it more clear in the updated paper which functional network the voxels we use belong to in Section 2.1 and throughout the results section.
>
> **W4**: We will note this in our updated paper. To be clear, the extent to which Kauf et al 2023 examines contiguous splits is a single bar in a supplemental figure showing decreased performance relative to shuffled splits. We maintain that our position on shuffled train-test splits is novel relative to Kauf et. al 2023. Kauf et. al 2023 argue that there are benefits to using shuffled train-test splits, namely better semantic coverage, and that the downsides of shuffled train-test splits can be addressed by decontextualizing input to the LLM (i.e. feeding each sentence in Pereira in isolation to the LLM). While it is true that shuffling can lead to better semantic coverage, passages in Pereira are explicitly designed such that there are 3-4 passages per semantic category. In our study, we leveraged this fact to design train-test splits in such a manner where sentences from only one passage from each semantic category were in the test set, allowing for strong coverage of the semantic space in the training set. We expand on why decontextualizing the input to the LLM is not a viable solution in **Q2**. We will include a discussion of how our work relates to Kauf et al. 2023 in our paper.
>
> **Q1**: Please let us know if we misinterpreted your comment, but we believe you may be interpreting the two glass brain plots on the left side of Figure 2d as SP+SL*, and the two glass brain plots on the right side of Figure 2d as SP+SL+GPT-XLU*. This may be because our figure legend was not clear enough, which we will make sure to fix. What we meant by the figure legend is that SP+SL* is on the left, and SP+SL+GPT2-XLU* is on the right, but they are placed right next to each other for each experiment. In this case, SP+SL+GPT-XLU* and SP+SL* perform similarly in the language network, as well as the auditory, visual, multiple demand, and default mode network (DMN). Aside from Figures 2/3d, all other figures in the main text (including Figures 2/3b) only include the language network. The voxels below the diagonal in Figure 2b were not significantly below (i.e. not better predicted when adding GPT2-XLU) and were distributed fairly uniformly across the language network. We will include a glass brain plot showing the difference between SP+SL+GLT2-XLU* and SP+SL* in our updated draft to make this clear.
>
> **Q2**: Kauf et al. (2023) used decontextualization (feeding in only the current sentence) to control for data leakage. However, this approach faces issues due to stimulus-independent neural activity. Temporally adjacent sentences tend to be more semantically similar than distant ones, so a trained LLM would likely represent sentences within the same passage similarly even in the decontextualized case, resembling OASM's representational structure. When using shuffled splits, this model could perform well even if neural responses are unrelated to the sentences presented. For example, if a participant thought about pizza throughout a passage about buildings, the linear regression would map the model representations from that passage to neural activity representing pizza, and would effectively predict pizza-related activity in the test set, yielding a high brain score. This issue persists even if the participant is attentive due to the ubiquity of stimulus-independent neural activity: if activity in a voxel deviates from its mean activity in a consistent manner throughout a passage for any reason, stimulus-driven or not, such a model could predict this deviation. The only robust solution we see to this problem is to hold-out whole passages in Pereira. Fortunately, *Pereira* is amenable to this design choice due to the semantic coverage described in W4. As an additional point, it is not at all clear how decontextualization could be done for a dataset like \it{Blank} or \it{Fedorenko}. To drive home these points, we  show that OASM explains a large portion of the neural variance explained by decontextualized GPT2-XL (GPT2-XL-DCTx) in Figure 3 of our PDF. Specifically, $\Omega_{\textit{GPT2-XL-DCTx}}(\textit{OASM})=$ $83.7 \pm 5.4$% in EXP1, and $82.5 \pm 5.2$% in EXP2, meaning OASM explains the overwhelming majority of neural variance that decontextualized GPT2-XL explains in *Pereira*. For this large fraction that OASM explains, it is impossible to determine whether the predicted neural responses are actually stimulus-driven.
>
> **Q3**:  Yes, the correction is designed to account for decreases in test performance from additional predictors due to overfitting on the training set. Such overfitting inevitably occurs in some voxels, and we wanted to minimize the extent to which it would artificially decrease the gain in variance explained when incorporating an LLM into the predictors.

---

> ### Comment · Reviewer_VpMi · 2024-08-11
>
> I thank the authors for responding to my comments.
>
> W2: Why are the pink/yellow bars so low for "LR, LT, Untrained, Pearson"? There appears to be quite big difference to BR, as well as SP? Can you please plot the scores as scatters against the R2 scores you report in the paper? It is challenging to see how much scores align with looking at different bar plots.
>
> W3: Thanks, this should be much clearer throughout the paper, and statistics/plots should consistently be reported for language-responsive or not (or at least make it very apparent what is what).
>
> Q1: Thanks for noting the confusion with the legends. However, I don't think my original question was answered with respect to visual regions? And for the dots below the diagonal in Figure 2b, how was it determined that they were not significant? And the fact that they were in the language network (and not elsewhere) makes sense? Because again, then visual regions might be driving the statistics? (this relates a bit to W3 where it is hard to make sense of which scores are reported for which voxels).
>
> Q2: I am not sure my original question was answered (sorry if I was unclear). But ultimately, doesn't OASM correspond to a predictor that has block-like structure, which is also the case when you have contextualized LLM representations in a block?
>
> Q3: Can you please clarify this?
>
> Other comments on global rebuttal:
>
> 1: Please correct me if I am reading Table 1 wrong, but GPT2-XL contributes a solid amount of neural variance?
>
> 2: Regarding your comment that "Finally, we used sum pooling because we found that LLM brain scores were in general higher when using sum pooling as compared to last token pooling (particularly so for untrained models), and we aimed to use methods that allow the LLM to perform maximally to avoid making our goal artificially easy. Furthermore, we disagree with the notion that using the last token representation is inherently more principled than sum-pooling. For each fMRI TR, the corresponding sentence was read between the 8th and 4th seconds preceding the TR. If the participant reads each sentence one word at a time, and each word induces a neural response, then the recorded image would show an HRF convolved with the neural responses to each word, roughly equal to the sum of the neural responses to each word.". Yes, I agree, but that *only* holds for Blank. The Pereira dataset was presented with a sentence on the screen, one at a time (full sentence, not word as you mention), and hence the logic you argue for does not hold -- and last-token pooling could be argued to be more suitable.
>
> 3: Per the global rebuttal and other reviewers comments on reproducing Schrimpf et al 2021, are you intending on changing the paper title?

---

> ### Author Response · Authors · 2024-08-12
>
> Thank you for the additional responses and questions.
>
> W2:
> Thank you for pointing this out. As we outlined in the detailed expansion section in the global response, we are working with “very ill-conditioned regressions due to high dimensional feature spaces, relatively few samples compared to the number of features, and noisy targets”. Vanilla linear regression (LR) is ill-suited in these cases, resulting in overfitting and leading to poor cross-validated correlations in contiguous train-test splits. This is additional evidence that BR is better suited for these cases. As for why LT doesn’t perform as well as SP, we speculate that this is because the sentence length (number of words/tokens) is not as linearly decodable from intermediate layers with LT as it is with SP. We explain why sentence length is linearly decodable from the intermediate layers when using SP in the first paragraph of *Section 3.2*, and provide a more formal proof of this in *Section A.9*. In summary, the fact that we are working with ill-conditioned regressions as well as the linear decodability of sentence length from intermediate layers with SP are explanations for why  the pink/yellow bars are so low for “LR, LT, Untrained, Pearson”. We will make sure to include this explanation in our updated draft.
>
> We will include  the scatter plot in the camera-ready version of the paper to facilitate easier comparison. Thank you for this feedback.
>
> W3:
> Thank you, we will make it more clear which statistics/plots correspond to which functional networks in our updated draft. To clarify for now, aside from Figures 2d and 3d in the main paper (and Figures 5 and 6d in the Appendix), every plot corresponds exclusively to language network voxels/electrodes/fROIs (as defined on the basis of their language-responsiveness by the Fedorenko lab and described further in the *Functional Localization* paragraph of **Section A.1**).
>
> Q1:
> Thank you for these comments.
>
> Regarding performance for the visual regions, we neglected to mention that the glass brain plots for each experiment come from a single representative participant. We will make sure to include this information in our updated draft. The result you mention where SP+SL predicts visual regions well was not robust across participants, which can be seen in the EXP2 plots on the right (where there is much higher predictivity in language network voxels). We will make sure to include glass brain plots for all participants in the Appendix to emphasize this. Also to clarify, the main focus of our glass brain plots in Figure 2d was to show that SP+SL* and SP+SL+GPT2-XLU* perform virtually identically across functional networks, including the language network. We will make sure to include an expanded discussion of our results on the glass brain plots in the updated draft.
>
> Regarding significance testing, we state in the results section that “we performed a one-sided t-test between the squared error values obtained over sentences between SP+SL+GPT2-XLU and SP+SL”. We provide a justification for this statistical test in **Section A.4**. As we describe in **Section 3.2**, lines 223-227, we then perform FDR correction within each participant and functional network (e.g. language network). This corrects for multiple comparisons, while still minimizing the number of false negatives since the number of tests in a given FDR correction is only the number of voxels in a single participant’s functional network, rather than the total number of voxels across all functional networks and/or all participants.
>
> We would like to clarify that it is not possible for visual regions to be driving the statistics, because we perform voxel-wise significance tests. Specifically, we find that “across all functional networks, only 1.26% (EXP1) and 1.42% (EXP2) of voxels were significantly (α = 0.05) better explained by the GPT2-XLU model before false discovery rate (FDR) correction; these numbers dropped to 0.001% (EXP1) and 0.078% (EXP2) after performing FDR correction within each participant and network [31]. None of the significant voxels after FDR correction were inside the language network. Taken together, these results suggest GPT2-XLU does not enhance neural prediction performance over sentence length and position even at the voxel level.” Since FDR correction is done within the language network within each participant, there is no influence of visual area voxels on our finding that there were no significant voxels within the language network.
>
> Lastly, although there were voxels where SP+SL+GPT-XLU performs better than SP+SL in the language network as you mention, none of these were significant. In other words, GPT2-XLU does not explain significant neural variance over SP+SL specifically in the language network. Please let us know if we have not addressed any of your questions, we are happy to provide additional insight.

---

> ### Author Response · Authors · 2024-08-12
>
> Q2:
> Yes, OASM does indeed correspond to a predictor that has block-like structure (specifically with temporally closer samples within a block being represented more similarly). It is true that such structure can also be present for contextualized (or decontextualized) LLM representations. However, such structure is also expected for the stimulus-independent (i.e. “noisy”) component of neural responses. The main message made in our rebuttal to this point is that, when using shuffled train-test splits, it is unclear whether the reported neural predictivity is due to predicting stimulus-dependent activity or stimulus-independent autocorrelated noise. Our additional variance partitioning analyses (Figure 3 in the new pdf) indicated that even when using decontextualized inputs for the LLM, the vast majority of the neural predictivity of the LLM could be accounted for by OASM. Importantly, because both stimulus-dependent and independent activity would be expected to have this OASM-like structure in Pereira, the fraction of neural responses predicted by GPT2-XL that can also be predicted by OASM cannot be disambiguated as stimulus-dependent or independent when using shuffled train-test splits. We will explain this clearly in our updated draft. Please let us know if we can clarify this point any further.
>
> Q3: Thank you for giving us the opportunity to clarify our voxel-wise correction approach. The core motivation is that we never want the incorporation of our simple features to bring down the reported performance of a model that includes an LLM as feature space. Hence, when using the voxel-wise correction for a model that includes an LLM as a feature space, the R2 taken for each voxel is that of the maximally performing sub-model for that voxel that includes the LLM, including the model that is just the LLM itself. To make the comparison fair, we do a similar procedure for the simple (non-LLM containing model), wherein we choose the best performing sub-model that does not include the LLM.
>
> For example, when comparing SP+SL* and SP+SL+GPT2-XLU*, we use the R2 of the best performing submodel in the set {SP, SL, SP+SL} for each voxel to compute SP+SL*, and we use the R2 of the best performing submodel in the set {GPT2-XLU, SP+GPT2-XLU, SL+GPT2-XLU, SP+SL+GPT2-XLU} for each voxel to compute SP+SL+GPT2-XLU*. We hope this clarifies any confusion, and are happy to answer any additional questions.
>
> Comment 1:
> It depends on how one defines a "solid amount". The main takeaway of Table 1 is that 81.2% of the variance explained by GPT2-XL can be explained by SP+SL+WORD in EXP1 and 90.1% of the variance explained by GPT2-XL can be explained by SP+SL+WORD in EXP2. Hence, the vast majority of variance explained by GPT2-XL here does not require contextual explanations. When further incorporating fairly limited contextual features (SENSE and SYNT), we explain 89.9% of variance explained by GPT2-XL in EXP1 and 97.8% in EXP2. Please let us know if we can make Table 1 more clear.
>
> Comment 2:
> We would like to clarify our argument regarding sum token pooling: we didn’t mean to imply that the sentence is presented one word at a time, rather that the participant “reads each sentence one word at a time,” and we apologize for the confusion. We assure you we appreciate your comments on the last token pooling, and we believe it is important to show that our results are consistent with both last token and sum pooling. To this end, we have shown that our results are consistent when using both last token and sum pooling methods in Pereira.  We will ensure that our updated draft places a larger focus on the last token pooling method than the current draft.
>
> Comment 3:
> We are open to modifying the title of our paper. Based on our understanding, the original criticism of our title was that we did not use the design choices used in Schrimpf et. al 2021. In our response, we have shown that our key findings on Pereira hold when using these design choices. If there are additional concerns regarding the title, we are happy to understand them so we can be as thoughtful as possible in the title.

---

> > ### Comment · Reviewer_VpMi · 2024-08-13
> >
> > I thank the authors for the additional clarifications and answers to my questions.

---

### Official Review · Reviewer_5zfv · 2024-07-04

**Soundness:** 3
**Presentation:** 2
**Contribution:** 3
**Rating:** 7
**Confidence:** 2

**Summary:**

This paper paper studies the topic of neural - brain representation mappings. They focus on three neural datasetse commonly used in LLM-to-brain mapping studies: Pereira fMRI, EcoG and Blank fMRI. Specifically, the study investigates the assumptions underpinning previous positive reports about the existence of mappings between brain representations and LLM internal representations. The study focuses in particular on GPT2-XL, which was shown to perform well on the Pereira dataset in particular, with which a series of brain-activation prediction experiments are performed.

The first presented result is that when shuffled train-test splits are used, the result is very different than when contiguous train-test splits are used, with opposite patterns on which layer performs best. This is particularly true for fMRI datasets. The authors then train an orthogonal auto-correlated sequences model on the shuffled split, which out-performs GPT-2-XL despite having a completely non-linguistic feature space. The authors take this as a signal that previous results should be challenged on their conclusion that high performance on this benchmark should be taken as an indication of brain-likeness.

Next, the authors investigate what explains the neural predictivity of an untrained GPT2-XL model, and they fi
nd that it is fully accounted for by sequence length and position. Following-up on that, they find that these
features are also main drivers for much of the neural predictivity of a trained GPT2-XL model.

**Strengths:**

- This paper presents a detailed study into why LLM activities may be predictive of neural activities, 'debunking' several previous claims. I think there is a lot of value in this
- The experiments seem sound (though I am not an expert in this field)
- The conclusions are interesting, and contain valuable lessons for future work on this topic

**Weaknesses:**

- The presentation could be improved, in my opinion. I don't always find everything completely clear. For instance, the notion of 'shuffled train-test splits' is quite important for the paper, but it is never really explained how they are specifically constructed, and how they differ from their 'contiguous' counterpart. (I can imagine multiple dimension in which one could shuffle)

Presentation suggestion: I think it may work better if the results of the different datasets are grouped together, result-by-result, rather than dataset by dataset.

**Questions:**

- It is not entirely clear to me *why* shuffled train-test splits are not good, could you explain that to me in non-technical terms?

**Limitations:**

The authors adequately address limitations.

---

> ### Author Rebuttal · Authors · 2024-08-07
>
> **W1**: Thank you for pointing this out, we will make sure to expand on our description of how shuffled train-test splits are constructed in our updated draft. We include this expanded description below:
>
> **Expanded description of shuffled train-test splits:** In neural encoding studies, datasets are typically of shape number-of-samples x number-of-voxels. Focusing on *Pereira*, each sample refers to one fMRI scan taken at the end of a sentence, and sentences are organized into passages of length 3-4 sentences. When using shuffled train-test splits, the data is randomly divided into training and testing sets along the number-of-samples dimension. For instance for *Pereira*, individual sentences are randomly selected and placed in the testing set. This means that it is possible for the second sentence in a passage to be placed in the testing set, and the first and third sentence within the same passage to be placed in the training set. By contrast, when using contiguous train-test splits with *Pereira*, an entire passage is either in the training or testing set, and sentences within a passage cannot both be within the training and testing sets.
>
> Thank you for the presentation suggestion.  Based on your remarks, we will incorporate summaries of results across datasets to help clarify the conclusions of the paper.
>
> **Q1**: An assumption implicit in language encoding studies using LLMs is that if the LLM exhibits high predictivity of brain responses, then it represents linguistic features of the stimuli that are also encoded in brain responses (e.g. features related to syntax or semantics). This assumption breaks down when using shuffled train-test splits. To see why, suppose we construct a very simple model which encodes each sentence as the average of sentences directly adjacent to it. Therefore, this simple model would represent the second sentence in a passage as the average of the first and third sentences in the passage. If the second sentence in a passage is in the testing set, and the first and third sentence in the same passage are in the training set, a regression fit using representations from this simple model would predict the brain response to the second sentence as the average of the brain responses to the first and third sentences. Since sentences within a passage are semantically similar and fMRI activity is autocorrelated in time, this prediction will perform quite well!
> To illustrate this intuition more concretely, suppose we constructed a trivial model termed the orthogonal autocorrelated sequence model (OASM). In *Pereira*, OASM relies on two principles: **1)** sentences from different passages are represented distinctly, and **2)** sentences within the same passage are represented similarly, such that sentences nearby in time within a passage are more similar than sentences further in time within a passage. Notably, these two principles do not reflect any deep aspect of human language processing, they simply allow for within-passage interpolation of brain responses. We describe OASM in more detail below in very simple terms, but this description can be skipped without loss of overall understanding.
>
> Our brain response matrix, $y$, is of shape number of samples ($N$) by number of voxels, where each sample is an fMRI image of the brain’s response following one sentence. The samples in $y$ are ordered such that sentences within the same passage are adjacent to each other and placed in the order in which they were presented. We proceed as follows:
>
> **1**: Construct an identity matrix of shape $N$ $\times$ $N$. An identity matrix is a matrix where all values are $0$ except along the diagonal, where the values are $1$.
>
> **2**: Apply a Gaussian blur within the blocks along the diagonal corresponding to each passage. This makes it so that the representations of sentences within a passage are similar, with sentences nearby in time being more similar. Sentences across passages remain orthogonal, in other words they contain no overlapping features.
>
> In sum, OASM is a blurred identity matrix with passage-level structure. We compared OASM to GPT2-XL, which was previously found to predict “$100%$ of explainable variance” in fMRI responses in Pereira. We found that OASM outperforms GPT2-XL, explaining around $50\%$more variance in Pereira when using shuffled train-test splits.
> If a model like OASM outperforms GPT2-XL with shuffled train-test splits, this calls into question previous results obtained using shuffled train-test splits. This is because some neural encoding studies quantify how “brain-like” a model is by how well it predicts neural activity. However, according to this logic, OASM is more brain-like than GPT2-XL when using shuffled train-test splits. Therefore, the use of neural encoding performance as a metric for brain similarity is not valid when using shuffled train-test splits.

---

> > ### Comment · Reviewer_5zfv · 2024-08-10
> >
> > Alright, thanks for your responses!

---

### Official Review · Reviewer_UQa9 · 2024-07-12

**Soundness:** 2
**Presentation:** 2
**Contribution:** 2
**Rating:** 3
**Confidence:** 4

**Summary:**

There is a large body of research focused on measuring the similarity between language processing in the brain and in language models. Recent studies have shown that representations from Transformer-based language models exhibit a higher degree of alignment with brain activity in language regions. However, the authors mention that this inference is valid only for the subset of neural activity predicted by large language models (LLMs).
The primary aim of this paper is to investigate this question by analyzing three popular neural datasets: Pereira, Blank, and Fedorenko. To achieve this, the authors build voxel-wise encoding models to compare encoding performance between representations from language models and brain recordings in three settings: (i) shuffled train-test splits during voxel-wise encoding, (ii) untrained LLM representations and their alignment with the brain, and (iii) trained LLM representations and their alignment with the brain. The experimental results demonstrate that untrained language models are explained by simple linguistic features such as sentence length and position, while trained language models are explained by non-contextual features (i.e., word embeddings).

**Strengths:**

1. This study primarily focuses on understanding the reasons behind the better alignment between language model representations and brain recordings. The exploration of various simple linguistic and non-contextual features, and their contribution to explaining the variance in brain predictivity over contextual embeddings, is valuable for the research community.
2. The authors tested different validation setups, including comparisons between untrained versus trained models and shuffled versus unshuffled data, to evaluate brain scores.
3. Controlling features with different combinations provided valuable insights into the contribution of each feature to the performance of brain alignment.

**Weaknesses:**

1. While the main research question aims to investigate the simplest set of features that account for the greatest portion of the mapping between LLMs and brain activity, the insights remain unclear for specific language regions of the brain. For instance, considering language parcels based on the Fedorenko lab, do simple features explain all the variance in these language regions? Or do these features only account for early sensory processing regions?
2. It is a well-known fact that Transformer-based representations consist of both low-level and high-level abstract features. If embeddings from language models predict brain activity and this predictivity is only due to a simple set of features, it should be better interpreted using approaches like residual analysis (Toneva et al. 2022), variance partitioning (Deniz et al. 2019), or indirect methods as suggested by Schrimpf et al. (2021).
3. Shuffling train-test splits is not an ideal scenario for brain encoding, especially for continuous language. All prior studies follow unshuffled train-test splits, i.e., contiguous time points (TRs). Shuffling the train-test split can result in sentences from the same passage being present in both the train and test sets, which is not ideal for model validation.
4. What are the implications of this study for both the AI and Neuroscience communities? What are the final conclusions?

**Questions:**

1. The experimental setup is weak and hard to follow:
- Why do authors consider R2 as evaluation metric instead of normalized brain alignment? Since authors used same 3 datasets from Schrimpf et al. 2021, it is a good comparison to use normalized brain alignment for percentage of expalined varinace.
- Since Reddy et al. (2021) performed both bootstrap and pairwise statistical tests to identify significant voxels, did the authors of this study also perform pairwise statistical significance tests to obtain corrected voxels across all the models?
- Why did the authors choose addition to obtain the sentence embeddings instead of averaging the word embeddings in a sentence? Additionally, how would considering the last token as the contextual representation of the sentence compare to these methods?
- All the datasets used in this study are based on reading and more of simple sentences. Did authors explored narrative datasets such as Harry Potter, Narratives, Moth-Radio-Hour, etc,.
- In Figure 3, the legend for the right plot is not clear. Additionally, the main takeaways from Figure 3 are difficult to follow.

**Limitations:**

Yes, the authors have presented several limitations in the conclusion. However, these limitations do not have any societal impacts on this work.

---

> ### Author Rebuttal · Authors · 2024-08-07
>
> **W1**: All our fMRI analyses on *Pereira* and *Blank*, with the exception of the glass brain plots in Figure 2d and 3d, are focused on the language network as defined by the same procedure used by *Fedorenko*. Since the language network is defined by selecting voxels that are more selective to sentences than non-words, it likely does not contain early sensory processing regions. In Schrimpf et. al 2021, they state the language network localizer “targets brain areas that support ‘high-level’ linguistic processing, past the perceptual (auditory/visual) analysis”. Our results on *Fedorenko* focus on the same electrodes used in the original Schrimpf et. al, 2021 analysis, which are language-responsive electrodes. We noted these details in **Section 2.1** for *Pereira* and *Fedorenko* datasets, but neglected to mention it for *Blank*. We apologize for this and will remedy this mistake. If the term “parcels” refers to subdivisions of the language network, we did not do this because our goal was to explain previous LLM-to-brain mappings which averaged results across the language network as a whole.
>
> **W2**: We agree that a variance partitioning framework is suitable for our work. We have updated our draft to include a variance partitioning metric, and we discuss it in depth in the global rebuttal and include results for *Pereira* in Table 1 of the PDF.
>
> **W3**: Yes we completely agree! Unfortunately, the comment that “all prior studies follow unshuffled train-test splits” is not correct. As we cite in our study, many previous papers have used shuffled train-test splits (Schrimpf et al. 2021, Kauf et al 2023, Hosseini et al 2023, ...). Kauf et al 2023 went as far as to defend the use of shuffled train-test splits, arguing that in spite of the data leakage, they allow for greater semantic coverage in the training data. We observed a paper published in the last month (AlKhamissi et al 2024) that continues to use shuffled train-test splits. We believe this is a persistent issue in the field that our paper addresses. A core contribution of our paper is that prior studies should use contiguous train-test splits. Shuffled train-test splits overestimate brain scores and result in different conclusions (Figure 1, Section 3.1 of our paper). We use contiguous train-test splits for the key conclusions of our paper.
>
> **W4**: There are three main implications of this study. First, shuffled train-test splits severely comprise neural encoding analyses. Although one might hope that this would be common knowledge, recent papers have argued that shuffled train-test splits are valid (Kauf et. al 2023), and we argue against this. Second, untrained LLM brain scores in these datasets can be explained by trivial features such as sentence length and positional signals. This is an important finding because Schrimpf et. al 2021 interpret untrained LLM brain scores as evidence that the transformer architecture biases computations to be more brain-like, without any analysis of what features account for the untrained model mapping. Finally, even trained LLM neural predictivity in these datasets can be largely accounted for by simple features.
>
> **Q1**: We have replicated our key findings in the same manner as in Schrimpf et al. 2021 (see Figure 1 in the rebuttal pdf) and found that our results are robust across a battery of design choices. To be clear, the “normalized brain alignment” method of Schrimpf et al. 2021 consists of training a non-regularized linear regression between the model features and neural responses, taking the pearson correlation on held out folds of the data, averaging the pearson correlation across folds per voxel, then taking the median pearson correlation across voxels within each participant. Though Schrimpf et al. 2021 also divided the brain scores by a noise ceiling, we do not because our goal is to examine the fraction of LLM brain score that simple features explain, and dividing by a noise ceiling would only scale, but not change, this result. This is because the noise ceiling division would be applied to both the LLM brain score and the simple feature set brain score, resulting in no change in the ratio between the two brain scores. We focus our paper on reporting out of sample R2 across voxels, with R2 values clipped at $0$ to prevent voxels that are noisily predicted at below chance accuracy from obscuring good performance on predictable voxels. We provide a description for why we use R2 in the expanded description section of the global response.
>
> **Q2**: We did not perform the bootstrapping procedure outlined in Reddy et. al 2021. Instead, we perform a *t*-test between the squared error values from two different models to test if one model has a higher R2 than another. We provide a justification for using this test in **Section A.4** in our paper.
>
> **Q3**: We tried average pooling and found that sum pooling works better. We can include average pooling results in our updated manuscript. We replicated our key findings on *Pereira* with last token pooling (Figure 1, 2 PDF).
>
> **Q4**: We did not explore more datasets, as we mention in our Conclusion. We include results from three neural datasets, and applying our framework to datasets such as from Lebel et. at 2023 would be outside the scope of our paper. Also, although we did not find interesting effects in *Blank*, *Blank* is a naturalistic dataset where participants listen to stories.
>
> **Q5**: Please let us know what specific aspects of Figure 3 you would like us to revise/provide clarification on. The main takeaway from Figure 3 is that a simple model, name SP+SL+WORD, which does not incorporate any form of contextual processing, accounts for a large amount of neural variance explained by GPT2-XL in *Pereira*. More complex features such as sense embeddings and syntactic representations account for an additional, but small portion of variance.

---

> > ### Comment · Reviewer_UQa9 · 2024-08-12
> >
> > Thank you, authors, for the clarification on several points. However, I still find some key conclusions of this paper unconvincing or uninteresting.
> >
> > 1. The statement that "Shuffled train-test splits overestimate brain scores and result in different conclusions (Figure 1, Section 3.1 of our paper)" is not surprising.
> >    - In shuffled train-test splits, there is a potential for a 'clock' (temporal) relationship, which might lead to information leakage during inference. This is why studies using naturalistic narrative datasets typically follow contiguous TRs. Therefore, the preference for contiguous train-test splits over shuffled train-test splits is an expected result.
> >
> > 2. The authors mentioned that they did not perform the bootstrapping procedure outlined in Reddy et al. 2021, but instead conducted a t-test between the squared error values from two different models to test if one model has a higher R² than the other.
> >    - When using the t-test to compare performance differences, how do the authors account for multiple comparisons or the potential dependency between test conditions?
> > 3. The observation that untrained LLM brain scores in these datasets can be explained by trivial features such as sentence length and positional signals is insightful.
> >    - However, this observation warrants a more detailed analysis. If all the variance is explained by trivial features alone, does that imply that only the lower layers of the untrained model are predicting brain signals well? What about the features from later layers? Additionally, if using untrained model features to predict early sensory regions, do we observe similar levels of alignment? Does this alignment also depend on simple features? More exploration in this area would be valuable.
> > 4. Schrimpf et al. 2021 also divided the brain scores by a noise ceiling, we do not because our goal is to examine the fraction of LLM brain score that simple features explain, and dividing by a noise ceiling would only scale, but not change, this result.
> >    -  Why is there no change in normalized brain predictivity? In both cases, whether using model predictions or trivial feature predictions, the results are always divided by the noise ceiling estimate. A more detailed explanation is needed to clarify why this normalization doesn't result in any observable change.
> > 5. Variance partitioning metric:
> >    - A Venn diagram with shared and unshared variance would provide a clearer understanding of the relationships, rather than simply presenting the information as numbers.
> > 6. Thank you for providing the additional results on sum pooling versus average pooling. This clarification addresses my question.

---

> ### Author Response · Authors · 2024-08-12
>
> Thank you for the additional responses and questions.
>
> 1. We personally agree that this is not surprising, but this is not the prevailing practice in neural encoding studies with these specific, widely used datasets. The novelty of our finding is not that this is “surprising”, but rather that it questions the interpretation of several moderate to highly impactful papers in the field. For instance, according to Google Scholar, Schrimpf et. al 2021 has been cited over 500 times in the past three years. Furthermore, studies in the field are continuing to use shuffled train-test splits (AlKhamissi et al. 2024), and recent studies have argued that shuffled train-test splits can be appropriate (Kauf et al. 2023). To reiterate, we are not stating that the use of contiguous train-test splits is novel, we are calling into question results from several previous papers using these neural datasets that have used shuffled train-test splits. We believe it is important to bring light to this issue because many readers are not aware that these studies employ shuffled train-test splits. We believe the continued use of shuffled train-test splits is harmful for the field, and that it is critical to address. Our goal is that the rest of the field, like you, would also find this not surprising, but we find this must be done through rigorous analyses, especially as others argue that shuffled train-test splits can be valid (Kauf et al., 2023).
>
> 2. We are happy to include alternative statistical tests, such as the bootstrapping procedure outlined in Reddy et. al 2021, in our updated draft. As we clarify, our statistical testing design choices were made to be conservative in supporting our conclusions. Our guiding philosophy for the statistical tests was that we wanted to minimize the number of false negatives. In other words, we wanted to avoid cases where the statistical test falsely reported that a model containing GPT2-XLU as a feature space did not significantly outperform a model not containing GPT2-XLU as a feature space. This is because we wanted to be liberal when stating for how many voxels/electrodes/fROIs does GPT2-XLU explain significant variance over simple features (to avoid overstating the importance of simple features). In line with this, we accounted for multiple comparisons by performing FDR correction within each functional network for each participant separately in Pereira. This minimizes the number of false negatives since the number of tests in a given FDR correction is only the number of voxels/electrodes/fROIs in a single participant’s functional network, rather than the total number of tests across all functional networks and/or all participants. If by dependency between test conditions you mean the dependency between temporally adjacent samples in the test set, we address this concern in Section A.4. In brief, we do not account for this temporal dependency because it only increases the rate of false positives (i.e. makes it more likely for us to say that a model containing GPT2-XLU outperforms a sub-model that does not contain GPT2-XLU), and so the net impact is that we overestimate for how many voxels/electrodes/fROIs does GPT2-XLU explain significant additional variance over simple features. Since there are no voxels/electrodes/fROIs where GPT2-XLU explains significant variance over simple features, this turned out to not be a concern.

---

> ### Author Response · Authors · 2024-08-12
>
> 3.
> Thanks for raising these interesting points. We report the performance across all layers for GPT2-XLU when using the sum pooling method in Figure 5a, and find that while the lower layers of GPT2-XLU tend to perform better when sum pooling, it is not by a very large margin. Therefore, the fact that all variance is explained by trivial features alone does not imply that only lower layers of the untrained model predict brain signals well. If the reviewer is suggesting that later layers might in fact have more complex features driving the mapping and that this warrants more detailed analysis, we will repeat our analyses for all layers of GPT2-XLU in our updated draft. To clarify, we did not do this in our original draft because previous neural encoding studies, including Schrimpf et al. 2021, focus their analyses on the best-performing layer of the LLM and we wanted to remain consistent in this manner.
>
> As for predictions in sensory regions, no, we do not see similar levels of alignment when using untrained models to predict early sensory regions. This can be seen in Figure 5a for sum pooling by comparing the visual network (which we consider a sensory region since the participants are reading the sentence) performance to the language network performance (and also for last token pooling in Figure 5c).
>
> We observe that the alignment between GPT2-XLU and the brain is explained by simple features across all functional networks, including the visual network. We state in Section 3.2 that across all functional networks, 0.001% (EXP1) and 0.078% (EXP2) voxels were significantly better explained using SP+SL+GPT2-XLU than SP+SL, and we will report statistics specifically for each functional network in the updated draft. Furthermore, it is unclear whether this small fraction exists because we use a liberal statistical test, as we discuss in our response to Q2. We will update the manuscript to clarify that there is an extremely small fraction of voxels where GPT2-XLU explains additional neural variance over SP+SL across all functional networks, including sensory regions. If the reviewer still believes additional analyses are worth pursuing in light of this, please let us know.
>
> 4.
> To be clear, applying the noise ceiling does lead to an observable change for a single model. However, our focus here is on comparing the brain score values between two models (specifically between a model with an LLM as a feature space and a sub-model without an LLM as a feature space). In this case, computing a noise ceiling is redundant. Just to be perfectly unambiguous about what we exactly did, if the raw brain predictivity of model A is a, the raw brain predictivity of model B be b, and the noise ceiling is c, then the “normalized” brain predictivity for model A is a/c and for model B is b/c. When we quantify the performance of model A relative to model B, we compute a/b, which is unaffected by noise ceilings, (a/c) / (b/c) = a/b. This applies to all of our quantifications of relative performance, as well as to the variance partitioning metric described in the global response. We will clarify this in the manuscript.
>
> Also, as stated in our rebuttal to reviewer bjuX: We also note that noise ceilings, when calculated by predicting each participant’s voxels with all other participants’ voxels and then extrapolating the performance to infinitely many participants, typically underestimate how much variance can be explained in a dataset. For instance, in a more recent paper using the same analysis choices as Schrimpf et al. 2021, several LLMs exceed the “noise ceiling” in Pereira, often by a large margin (Aw et al. 2023). Lastly, computing the noise ceiling in the same style as in Schrimpf et al. 2021 is quite computationally expensive. For all of these reasons, we were unable to compute a noise ceiling prior to the end of the rebuttal period, but would be willing to include it in the camera ready version if desired.
>
> 5.
> Thank you for this visualization suggestion. We will add it to the manuscript.

---

> ### Comment · Reviewer_UQa9 · 2024-08-12
>
> Thank you, authors, for providing clarification on statistical tests.
>
> The early papers [Schrimpf et al. 2021], [Kauf et al. 2021] have introduced many novel findings. For instance, [Schrimpf et al. 2021] was the first study to reverse-engineer many language models as subjects to estimate brain alignment. This study explored 43 language models and introduced the concept of noise ceiling estimates. Similarly, [Kauf et al. 2021] focused on understanding the reasons for better alignment between language models and brain recordings, exploring various syntactic and semantic perturbations to verify these reasons.
>
> In contrast, the current study focuses on comparing shuffled train-test splits with contiguous train-test splits. What are the key takeaways for the computational cognitive neuroscience community? As it stands, the results are not particularly surprising. Therefore, the current version requires additional interpretation, particularly focusing on the analysis of simple features at different layers, as well as exploration using other narrative ecological datasets.

---

> > ### Author Response · Authors · 2024-08-13
> >
> > > The early papers [Schrimpf et al. 2021], [Kauf et al. 2021] have introduced many novel findings… Similarly, [Kauf et al. 2021] focused on understanding the reasons for better alignment between language models and brain recordings, exploring various syntactic and semantic perturbations to verify these reasons.
> >
> >
> > These findings were found using shuffled train-test splits, which you previously stated is “not ideal for brain encoding studies” because of “information leakage during inference.” You also agreed that this “can result in different conclusions (Figure 1, Section 3.1 of our paper)," saying this statement was “not surprising.” We are therefore genuinely confused, since it appears you are now comparing us with these studies whose conclusions use shuffled train-test splits, a concern we both agreed impacts their conclusions. We would appreciate any clarification.
> >
> >
> > > For instance, [Schrimpf et al. 2021] was the first study to reverse-engineer many language models as subjects to estimate brain alignment.
> >
> > We are uncertain what you mean by “reverse-engineer.” But we want to be sure the reviewer understands that our analyses showing that trained and untrained GPT brain scores can be explained by simpler features have not been done before. This was not present in Schrimpf et al., 2021 and we are not aware of any other study that does this.
> >
> >
> > > This study explored 43 language models and introduced the concept of noise ceiling estimates.
> >
> >
> > We would like to clarify to the reviewer that Schrimpf et al 2021 did not introduce the concept of noise ceiling estimates. The earliest use of noise normalization for neural predictivity metrics that we are aware of is Schoppe et al. 2016. It is unclear how the reviewer believes these points relate to our work.
> >
> > Schoppe O, Harper NS, Willmore BDB, King AJ and Schnupp JWH (2016) Measuring the Performance of Neural Models. Front. Comput. Neurosci. 10:10. doi: 10.3389/fncom.2016.00010
> >
> > > In contrast, the current study focuses on comparing shuffled train-test splits with contiguous train-test splits.
> >
> >
> > While our rebuttal period focused on train-test splits, this was to correct the reviewer’s false statement that “all prior studies follow unshuffled train-test splits” and further discuss its significant impact on prior work. We’d like to clarify that the train-test splits are only one conclusion from our paper, and certainly not the only focus. This result (**Section 3.1 and Figure 1**) only comprises ~1 page of our manuscript.
> >
> >
> > > What are the key takeaways for the computational cognitive neuroscience community?
> >
> >
> > We would like to note that we have already answered this in our response to **W4**, in our **Abstract**, throughout our paper, and in the Limitations and Conclusion section. However, we are happy to summarize our contributions here. In addition to the conclusion on train-test splits, we would like the computational cognitive neuroscience community to make these takeaways:
> >
> > Untrained GPT2-XL neural predictivity is fully accounted for by sentence length and position. To understand this result in more detail, please see results in **Section 3.2, Figure 2**, and our **Abstract**. This section is important because, as stated in our **Abstract**, it “undermines evidence used to claim that the transformer architecture biases computations to be more brain-like.”
> >
> > Sentence length, sentence position, and static embeddings account for the majority of trained GPT2-XL neural predictivity. Furthermore, sense embeddings and syntactic features account for a small, additional portion of trained GPT2-XL neural predictivity. To understand this result in more detail, please see results **Section 3.3, Figure 3, Table 1, Table 1 (in our review PDF), Appendix A.6** (for reproducing our findings with Roberta-Large), and our **Abstract**. This section is important because, as stated in our abstract, it helps us to “conclude that over-reliance on brain scores can lead to over-interpretations of similarity between LLMs and brains, and emphasize the importance of deconstructing what LLMs are mapping to in neural signals.”
> >
> > Please also know that **Section 4** (*Fedorenko*) and **Section 5** (*Blank*) contain results of these analyses on other datasets.

---

> > > ### Author Response · Authors · 2024-08-13
> > >
> > > > As it stands, the results are not particularly surprising.
> > >
> > >
> > > Perhaps we can agree that whether a finding is surprising or not is partly subjective. We noted earlier, you stated “the observation that untrained LLM brain scores in these datasets can be explained by trivial features such as sentence length and positional signals is insightful.” While we agree it is insightful, is it (e.g.) surprising that brain scores for untrained and trained GPT2-XL are mostly explained by simple, non-contextual features in *Pereira* and *Fedorenko*? Is it surprising that when using contiguous train-test splits on *Blank*, not a single fROI is predicted above chance by GPT2-XL? We believe so because it stands in contrast to a significant literature using these datasets, and perhaps it is therefore surprising to others in our field.
> > >
> > >
> > > > Therefore, the current version requires additional interpretation, particularly focusing on the analysis of simple features at different layers, as well as exploration using other narrative ecological datasets.
> > >
> > >
> > > Regarding the layer analysis, as we stated in our prior rebuttal, we will perform these relatively straightforward analyses for the camera ready version. Due to the timing of this new concern, we are unable to address it during the discussion period. Regarding narrative ecological datasets, we respectfully disagree with this being in the scope of our paper. The scope of our paper is to address three widely used datasets and some major missteps and over-interpretations made in previous analyses of them, and we provide a comprehensive analysis into what features account for untrained and trained GPT2-XL's neural predictivity on these datasets. As it is, our paper addresses three neural encoding datasets, more than most other neural encoding studies that we are aware of. We additionally already addressed this in our **Q4** response in our first rebuttal.

---

> > > > ### Comment · Reviewer_UQa9 · 2024-08-13
> > > >
> > > > Thank you, authors, for providing some clarification. I would be happy to update my score if the authors could clarify the following questions:
> > > >
> > > > Q1. Authors state that both trained and untrained GPT brain scores can be explained by simpler features have not been done before.
> > > >
> > > >  Using GPT model representations for brain-LM alignment has been extensively explored in various brain encoding studies:
> > > >
> > > >  [Antenolleo et al. 2021], Low-dimensional structure in the space of language representations is reflected in brain responses, NeurIPS-2021
> > > >
> > > >  [Caucheteux et al. 2021], Disentangling syntax and semantics in the brain with deep networks, ICML-2021
> > > >
> > > >  [Caucheteux et al. 2022], Deep language algorithms predict semantic comprehension from brain activity, Scientific Reports-2022
> > > >
> > > >  [Caucheteux et al. 2022], Brains and algorithms partially converge in natural language processing, Nature Communications Biology-2022
> > > >
> > > >  [Merlin et al. 2023], Language models and brain alignment: beyond word-level semantics and prediction, Arxiv-2022
> > > >
> > > >  [Oota et al. 2023], What aspects of NLP models and brain datasets affect brain-NLP alignment?, CCN-2023
> > > >
> > > >  [Oota et al. 2024], Speech language models lack important brain-relevant semantics, ACL-2024
> > > >
> > > > These studies have demonstrated that GPT-based language models predict brain activity across all language regions  to an impressive degree, regardless of whether the activity is text-evoked or speech-evoked. They all follow unshuffled train-test splits and consistently show better alignment than simpler word embedding features. Therefore, the statement that both trained and untrained GPT brain scores can be explained by simpler features raises questions about the findings of these studies, as they also utilize GPT-based models and report superior brain alignment for language regions. This is why I am particularly emphasizing this point. It would be helpful if the authors could share their views on this aspect.
> > > >
> > > > Q2. The earliest use of noise normalization for neural predictivity metrics that we are aware of is Schoppe et al. 2016.
> > > > * In the work by [Schrimpf et al. 2021], noise ceiling estimation is performed by using one subject as the target and the remaining subjects as the source.
> > > > * In contrast, [Schoppe et al. 2016] estimate the noise ceiling based on inter-trial repetitions. These approaches highlight a difference in how noise ceiling estimates are derived in both studies.

---

> ### Author Response · Authors · 2024-08-13
>
> Q1: Thank you for bringing up these other studies. We would like to clarify our response that “our analyses showing that trained and untrained GPT brain scores can be explained by simpler features have not been done before” was made in the context of discussing the novelty of our analyses relative to Schrimpf et. al 2021, and was made in regards to the specific datasets used in Schrimpf et. al 2021 and our study. Throughout our paper and in our responses, we are very careful to state that our findings apply to the specific datasets used in our study. In fact, we state in our Limitations and Conclusion that “we did not analyze datasets with large amounts of neural data per participant, for instance Lebel et. al 2023 in which the gap between the neural predictivity of simple and complex features might be much larger.” We would like to also emphasize that the datasets we address have been used in many studies that have used LLM-to-brain mappings to motivate neuroscientific conclusions (e.g. Schrimpf et al. 2021, Hosseini et al. 2023, Kauf et al. 2023, Aw et al. 2023, AlKhamissi et al. 2024). We apologize if this was ambiguous, and thank the reviewer for giving us the opportunity to further emphasize this point.
>
> Regarding the studies cited by the reviewer, we would like to emphasize the following points. **1)** Notably, none of these studies use any of the three neural datasets used in our study, and therefore we make no claims about the degree to which simple features account for LLM to brain mappings in the neural datasets employed in these studies. **2)** Based on our understanding, the only paper cited which discusses untrained LLM findings is “Brains and algorithms partially converge in natural language processing”. However, the goal of that study was not to examine what features drive the above chance neural predictivity of untrained LLMs, and so they did not perform the same analyses we did using simple features. **3)** Based on our understanding, the studies cited do not show on the datasets they use that such simple, non-contextual features explain such a large fraction of trained LLM brain scores (which we showed on Pereira and Fedorenko), or that when performing train-test splits correctly trained LLM brain scores fall to chance levels (which we showed on Blank). Therefore, we believe our findings on trained LLMs with the specific datasets we use are also novel. We leave it to future work to examine whether simpler explanations account for as much of the variance predicted by LLMs in other datasets.
>
> We will reference these papers in a related works section in the updated draft.
>
> Q2: Thank you for bringing up this distinction in how noise ceiling values are computed. In our initial response, we were referring to the concept of noise ceiling generally rather than the specific methods used to compute them. We would like to emphasize that noise ceiling computations do not impact our findings (see point 4 two rebuttals ago).

---

### Official Review · Reviewer_um4X · 2024-07-16

**Soundness:** 2
**Presentation:** 2
**Contribution:** 2
**Rating:** 4
**Confidence:** 4

**Summary:**

This paper investigates the similarity between large language models (LLMs) and human brain activity by analyzing brain scores, which measure how well a model predicts neural signals. The authors question the validity of using brain scores as a measure of similarity between LLMs and human cognitive processes. They analyze three neural datasets and find that simple features like sentence length and position explain much of the neural variance that LLMs account for. They caution against over-reliance on brain scores and emphasize the need for a detailed deconstruction of what LLMs are mapping to in neural signals.

**Strengths:**

The study provides a thorough examination of how various features (simple and complex) contribute to the neural predictivity of LLMs, offering a detailed deconstruction of the relationship between LLMs and brain activity. The replication of key findings using RoBERTa-Large, in addition to GPT2-XL, strengthens the validity of the conclusions drawn regarding the generalizability of the results across different LLM architectures

**Weaknesses:**

1.  The methodology and findings are not particularly novel. Previous studies have already suggested that untrained LLMs can achieve good brain scores and that sentence length and position are significant predictors. Thus, two of the three core contributions claimed by the authors are not unique to this paper.
2. While the authors conclude that over-reliance on brain scores can lead to over-interpretations of similarity between LLMs and brains, it is not clear how this conclusion is drawn from the experimental results. The study itself relies heavily on brain scores to make its arguments, and the authors do not explicitly state what aspects of previous work have been over-interpreted.

**Questions:**

1. How do the findings of this paper significantly advance our understanding beyond what has already been established about untrained LLMs and simple feature predictivity?
2. What specific previous findings are over-interpreted when relying on brain scores? How can over-reliance on brain scores be quantified or defined?
3. What alternative methods or interpretations do the authors propose for evaluating the alignment between LLMs and brain activity if the current reliance on brain scores is problematic?

**Limitations:**

Yes

---

> ### Author Rebuttal · Authors · 2024-08-07
>
> **W1**: We believe there may be a misunderstanding of our core findings. We clarify our three core contributions here as well as their novelty + uniqueness, which we state in our abstract as well as throughout our paper:
> **1**: Shuffled train-test splits, which have been used by multiple previous LLM-to-brain mapping studies, severely compromise brain score analyses. We thus use contiguous train-test splits for contributions 2 and 3.
> **2**: Untrained LLM brain scores can be fully accounted for by positional and sentence length signals on *Pereira* and *Fedorenko*, and are at chance levels in *Blank*.
> **3**: Trained LLM brain scores can be largely accounted for by positional signals, sentence length, and static word embeddings in *Pereira*, word position in *Fedorenko*, and are at chance levels in *Blank*.
>
> Importantly, our core contribution is *not* that untrained LLMs map well to brain activity (we cite four papers in our Introduction section which have already demonstrated this result)  or in using SP and SL, although we are not aware of previous work using these features in any LLM-to-brain mapping study. Rather, before our study it was largely a mystery why untrained LLM brain scores were so high. In fact, in Schrimpf et. al 2021, untrained GPT2-XL brain score was roughly double that of trained GPT2-XL brain score for Blank and about 75% as high as the trained GPT2-XL brain score for Pereira (see Figure S8 in their paper).
>
> The core contribution of our study regarding untrained LLMs and SP and SL is to demonstrate that 1) untrained LLM brain scores in Schrimpf et. al 2021 are artificially inflated due to shuffled train-test splits, and 2) when using contiguous train-test splits, SP and SL account for all neural variance explained by untrained GPT2-XL. This is an important finding because Schrimpf et. al 2021 interpret untrained LLM brain scores as evidence that the transformer architecture biases computations to be more brain-like. By contrast, our study shows that untrained LLM brain scores were significantly overstated in these datasets, and the remaining neural variance is explained by features not unique to the transformer architecture. We will make this clearer in our manuscript.
>
> **W2**: We clarify that our study is not advocating for the abandonment of brain score metrics. Rather, we are cautioning against the application of brain scores between complex models and neural activity without appropriate controls; we believe that reporting brain scores along with a rigorous examination of what drives the mapping between LLMs and the brain can be beneficial for the field. We use brain scores to show that simpler features can explain over-interpreted results (see our responses in W1, Q2). While we believe we stated over-interpreted results in our abstract, for example: “this undermines evidence used to claim that the transformer architecture biases computations to be more brain-like,” we will state this more explicitly in our manuscript, including the Conclusion.
>
> At a high level, we show that brain scores do not paint a full picture of correspondence between LLMs and the brain. This is because they can be artificially high due to incorrect methods, for instance shuffled train-test splits, or they can be driven by relatively simple features, as we show with both untrained and trained LLM brain scores. Our results clearly demonstrate how over-reliance on untrained LLM brain scores led to the over-interpretation in Schrimpf et. al 2021 that features unique to the transformer architecture bias computations to be more brain-like. More generally, over-reliance on brain scores has led to a narrative that LLMs serve as potential neurobiological models of language processing, and our study shows without rigorously accounting for what is driving brain scores, evidence supporting this narrative may be contaminated by factors like temporal autocorrelation from shuffled splits and features that are not unique to LLMs. We will make sure to expand on this in our updated draft.
>
> **Q1**: We are the first study, to our knowledge, to show that previously reported high correlations between untrained LLMs and the brain are explained by simple features not unique to LLMs. We believe this is an important advancement in the field of LLM-to-brain mappings because before our study it was a mystery why untrained LLMs map to brain activity at above chance levels.
>
> **Q2**: The specific over-interpretations that we address are: **1)** untrained LLM brain scores are evidence that the transformer architecture biases computations to be more brain-like, and **2)** the reason GPT2-XL outperforms other models is due to representations unique to auto-regressive LLMs (i.e. representations related to next-word prediction). Our findings push back on both of these claims by showing that simpler features can account for all of untrained GPT2-XL brain scores, and the majority of GPT2-XL brain scores are accounted for by non-contextual features. Over-reliance in the context of our paper can be defined as excessive reliance on a scalar metric for neural encoding, without a rigorous examination of what representations are driving this metric. Without such a rigorous examination, the representations driving the neural encoding performance may be theoretically uninteresting (e.g. sentence length and sentence position). We will make this more clear in our manuscript.
>
> **Q3**: The alternative that we propose is that researchers should be cautious about directly interpreting higher brain scores as evidence that one model is more brain-like than another, and instead they should seek to understand what specific aspects of a complex model drive high brain scores. Our paper demonstrates this approach, examining how much neural variance an LLM explains over simpler features.

---

> > ### Author Response · Authors · 2024-08-09
> > **Minor clarification on second core contribution**
> >
> > Sorry, we noticed that we did not explicitly state that neural predictivity of GPT2-XLU (the untrained model) was at chance levels on the Blank dataset when using contiguous splits in the submitted draft. This result has been incorporated explicitly in our updated draft.

---

### Author Rebuttal · Authors · 2024-08-07

We thank the reviewers for all the feedback. We have performed several new experiments to address the following concerns.

**1**: In our original draft, we departed from the design choices made in Schrimpf et. al 2021. Specifically, we used banded ridge regression instead of vanilla linear regression, sum pooling instead of last token pooling, and used out of sample R2 instead of Pearson correlation. To ensure our key findings were not driven by these distinctions, we replicated the following findings using linear regression, last token pooling, and Pearson correlation. **1)** When using shuffled train-test splits a trivial model, OASM, which only encodes within passage temporal auto-correlation outperforms GPT2-XL and accounts for the majority of the neural variance explained by GPT2-XL (Figure 2 in PDF). **2)** SP+SL accounts for all the neural predictivity of GPT2-XLU (Figure 1). **3)** SP+SL+WORD accounts for the majority of the neural predictivity of GPT2-XL (Figure 1). We compute Pearson correlation in the same manner as Schrimpf et. al 2021 by taking the mean correlation across folds for each voxel (without clipping negative values to zero), and then taking the median correlation across voxels in the language network for each participant. We also ensured that our key findings hold when performing other permutations of these design choices, such as using banded ridge regression with last token pooling and Pearson correlation, to ensure robustness across a variety of methods (Figure 1).  We provide an expanded discussion of why we departed from the design choices in Schrimpf et. al 2021 below.

**2**: Another piece of consistent feedback we received was that we did not report a variance partitioning metric, and so for instance it is unclear what fraction of GPT2-XL brain score OASM explains. We have therefore updated our draft to include a variance partitioning metric. Specifically, given a model $M$, we quantify the percentage of neural variance explained by an LLM that is also explained by $M$ as:

$\Omega_{\textit{LLM}}(\textit{M}) = \left(1 - \frac{R^2_{\textit{M}+\textbf{\textit{LLM}}}* - R^2_{\textit{M}}*}{R^2_{\textit{LLM}}}\right) \times 100\%$.

For a given participant, $\Omega_{\textit{LLM}}(\textit{M})$ is less than (more than) $100$% if, on average, the best sub-model that includes the LLM per each voxel/electrode/fROI outperforms (underperforms) the best sub-model that does not include the LLM per each voxel/electrode/fROI. When reporting this quantity, we clip the per participant values to be at most equal to $100$% to prevent noisy estimates from leading to an upward bias, and then report the mean $\pm$ standard error of the mean (SEM) across participants. We report these values in Table 1 in the PDF file for *Pereira* trained results and within reviewer rebuttal sections when relevant. All our conclusions are upheld under these analyses.

—-----------------

**Detailed expansion- Reasoning behind differences in design choices from Schrimpf et. al 2021:**  First, as is standard for neural encoding studies, we are generally working with very ill-conditioned regressions due to high dimensional feature spaces, relatively few samples compared to the number of features, and noisy targets. Because the ill-conditioning of standard linear regression disfavors large feature spaces relative to smaller ones, it generally results in an LLM appearing to explain less unique variance beyond our simple feature spaces than when ridge regression is used (as we show in Figure 1 in the PDF). Furthermore, the banded regression procedure we employ, which allows for fitting regressions with multiple feature spaces, depends on having a regularization parameter. In sum, our use of ridge regression is consistent with nearly all previous neural encoding studies that we are aware of (except for Schrimpf et. al 2021, Kauf et. al 2024, and Hosseini et. al 2024), is well-motivated due to the large number of features and different feature spaces, and only made it more difficult for us to account for the neural variance explained by GPT2-XL. Second, we did not use Pearson correlation because R2 can be interpreted as the fraction of variance explained, whereas Pearson correlation cannot. In our updated draft we include a metric which quantifies the fraction of LLM neural variance explained that simple features account for, and this would not be possible with Pearson correlation. Finally, we used sum pooling because we found that LLM brain scores were in general higher when using sum pooling as compared to last token pooling (particularly so for untrained models), and we aimed to use methods that allow the LLM to perform maximally to avoid making our goal artificially easy. Furthermore, we disagree with the notion that using the last token representation is inherently more principled than sum-pooling. For each fMRI TR, the corresponding sentence was read between the 8th and 4th seconds preceding the TR. If the participant reads each sentence one word at a time, and each word induces a neural response, then the recorded image would show an HRF convolved with the neural responses to each word, roughly equal to the sum of the neural responses to each word.

---

### Author Response · Authors · 2024-08-07
**References**

Throughout our rebuttals, we refer to the following works:

Schrimpf, M., Blank, I. A., Tuckute, G., Kauf, C., Hosseini, E. A., Kanwisher, N., Tenenbaum, J. B., & Fedorenko, E. The neural architecture of language: Integrative modeling converges on predictive processing.

Kauf C, Tuckute G, Levy R, Andreas J, Fedorenko E. Lexical semantic content, not syntactic structure, is the main contributor to ANN-brain similarity of fMRI responses in the language network

Hosseini EA, Schrimpf M, Zhang Y, Bowman S, Zaslavsky N, Fedorenko E. Artificial Neural Network Language Models Predict Human Brain Responses to Language Even After a Developmentally Realistic Amount of Training

Aw KL, Montariol S, AlKhamissi B, Schrimpf M, Bosselut A. Instruction-tuned LLMs with World Knowledge are More Aligned to the Human Brain

AlKhamissi, B., Tuckute, G., Bosselut, A., & Schrimpf, M. Brain-Like Language Processing via a Shallow Untrained Multihead Attention Network.

---

### Decision · Program_Chairs · 2024-09-25

**Decision:**

Reject

**Comment:**

This work investigates the reasons for observed alignment between untrained and trained LLMs and human brain recordings. There are 3 main findings: (1) a specific line of work that investigates brain-LLM alignment, which uses shuffled train-test splits, is overestimating the existing alignment; (2) brain-LLM alignment with untrained LLMs can be explained by simple linguistic features such as sentence length and position; (3) brain-LLM alignment with trained LLMs can be somewhat explained by word embeddings.

Studying the reasons behind brain-LLM alignment is important. However, there are significant concerns with the current version of the paper. The first is the framing of the work, which mostly concerns finding (1) above. Currently, the work reads as a general takedown of brain-LLM alignment. However, the investigated setting here of shuffled train-test splits is a very specific one, used by mostly one group. I agree that several of the cited papers that use this split are influential, but there has been a long line of work on brain-LM alignment, mostly using naturalistic data (as pointed out by Reviewer UQa9), that specifically uses contiguous test data exactly for the reasons that the authors explain. This fact needs to be made much more clear in a revised version of this work -- the fact that much of the field is doing things correctly should be acknowledged, so that not to disparage the entire subfield based on choices made by one group, however influential this group is.

The second major concern is novelty. (2) above is the most novel finding in the work, though as the authors themselves say, this work is not well positioned to be related to the findings from many other works, which use other naturalistic datasets. I suggest that the authors include results from at least one commonly used dataset for brain-LLM alignment in their revision -- Moth Radio Hour, Harry Potter, or Narratives -- so that the findings can be compared to those from other papers. Regarding (3), there are several works that have already shown that word embeddings have a big impact on brain-LLM alignment [Toneva et al. 2022 Nature Computational Science, Srikant et al. 2023 NeurIPS] so the manuscript needs to discuss the relationship to these works more in depth and to make it more clear what the current work is contributing over these existing works.